



# An assessment of geographical distribution of different plant functional types over North America simulated using the CLASS-CTEM modelling framework

**Rudra K. Shrestha[1], Vivek K. Arora[1], Joe R. Melton[2], and Laxmi Sushama[3]**

[1]Canadian Centre for Climate Modelling and Analysis, Environment and Climate Change Canada, University of Victoria, Victoria, BC, V8W 2Y2, Canada

[2]Climate Research Division, Environment and Climate Change Canada, Toronto, Ontario, Canada

[3]Département des sciences de la Terre et de l'atmosphère, Université du Québec à Montréal, Canada

*Correspondence to: V. K. Arora (Vivek.Arora@canada.ca)*

## Abstract

The performance of the competition module of the CLASS-CTEM (Canadian Land Surface Scheme and Canadian Terrestrial Ecosystem Model) modelling framework is assessed at 1° spatial resolution over North America by comparing the simulated geographical distribution of plant functional types (PFTs) with two observation-based estimates. The model successfully reproduces the broad geographical distribution of trees, grasses and bare ground although limitations remain. In particular, compared to the two observation-based estimates, the simulated fractional vegetation coverage is lower in the arid south-west North American region and higher in the Arctic region. The lower than observed simulated vegetation coverage in the south-west region is attributed to lack of representation of shrubs in the model and plausible errors in the observation-based data sets. The observation-based data indicates vegetation fractional coverage of more than 60% in this arid region, despite only 200-300 mm of precipitation that the region receives annually and observation-based leaf area index (LAI) in the region are lower than one. The higher than observed vegetation fractional coverage in the Arctic is due to the lack of representation of moss and lichen PFTs and also likely because of inadequate representation of permafrost in the model as a result of which the $C_3$ grass PFT performs overly well in the region. The model generally reproduces the broad spatial distribution and the total area covered by the two primary tree PFTs (needleleaf evergreen and broadleaf cold deciduous trees) reasonably



well. The simulated fractional coverage of tree PFTs increases after 1960s in response to the $CO_2$
fertilization effect and climate warming. Differences between observed and simulated PFT
coverages highlight limitations in the model and provide insight into physical and structural
processes that need improvement.



# 1 Introduction

The terrestrial ecosystem plays an important role in regulating climate and weather through land-atmosphere exchange of water and energy (Cramer et al., 2001; Garnaud et al., 2015; Pielke et al., 1998; Ran et al., 2016) and in mitigating climate change by sequestering atmospheric $CO_2$ (Bonan, 2008; Timmons et al., 2016). The projected sink of atmospheric $CO_2$ is uncertain due to disagreements among the Earth system models (ESMs) (Arora et al., 2013; Friedlingstein et al., 2006) primarily due to differing responses of their terrestrial ecosystem modules to future changes in atmospheric $CO_2$. This uncertainty arises primarily because of the differences in the strength of the $CO_2$ fertilization effect on the land carbon cycle components (Arora et al., 2013; Cramer et al., 2001; Friend et al., 2013) but also because of differences in the response of vegetation. Models differ in how the spatial distribution of vegetation, and its composition, changes in response to changing climate and increasing $CO_2$ (Cramer et al., 2001). These differences are also resolution dependent. For example, models with coarse grid resolutions cannot explicitly resolve climatic niches, which in turn potentially contributes to biases in simulated vegetation distribution (Melton and Arora, 2016; Shrestha et al., 2016).

Vegetation responds to changes in climate and atmospheric $CO_2$ concentration by changing its structural attributes including leaf area index (LAI), rooting depth, vegetation height, and canopy mass, as well as its areal extent. Structural vegetation changes generally occur over seasonal to decadal time scales (Kramer and Kozlowski, 1979), while the slower areal extent changes typically occur on decadal to centennial time scales (Ritchie and Macdonald, 1986). The dynamic behavior of vegetation affects weather and climate due to its strong control over biophysical processes. At hourly to daily timescales, vegetation affects the exchange of water and energy between the land surface and the atmosphere primarily through the control of leaf stomata. At longer timescales from seasonal, annual to decadal timescales, vegetation affects components of energy and water balance through its structure (LAI, rooting depth, etc.) and its areal extent and thereby land surface albedo. Conversely, dynamics of vegetation is directly influenced by climate and the competitive ability of the plants. In this way vegetation responds to climate by changing its structure and areal extent depending on the colonization ability of plants.



These climate-vegetation interactions have been well documented (e.g. Gobron et al., 2010;
Wang et al., 2011).

Natural vegetation is typically characterized in dynamic global vegetation models (DGVMs)
based on a limited number of PFTs (Sitch et al., 2003) because it is impossible to represent
thousands of species in a model. Species characterized by similar attributes, mainly based on
their form and interactions with the environment (Box, 1996), are grouped together as a single
PFT. For example, tree species with similar leaf form such as fir *(Abies)*, spruce *(Picea)* and pine
*(Pinus)* are classified as needleleaf evergreen trees. The geographical distribution of the PFTs in
DGVMs is determined by their ability to grow and increase their areal extent given certain
climate and soil conditions and their competitive ability.

One way of representing competition between PFTs in DGVMs is through the use of the Lotka-
Volterra (LV) equations. While originally developed for predator-prey competition, the LV
equations have been used in a number of DGVMs (Arora and Boer, 2006; Brentnall et al., 2005;
Cox, 2001; Zhang et al., 2015). The use of the classical form of the LV equations for modelling
competition between PFTs, however, leads to an amplified expression of dominance in that the
dominant PFT ends up occupying a disproportionately large fraction of a grid cell leading to
little co-existence between PFTs. Arora and Boer (2006) proposed changes to the classical
implementation of the LV equations for modelling competition between PFTs to reduce this
amplified expression of dominance. Their approach, which has been implemented in the CLASS-
CTEM modelling framework and which allows improved co-existence of PFTs compared to the
classical LV equations, has been shown to simulate vegetation distribution reasonably well at the
global  (Melton and Arora, 2016) as well as point (Shrestha et al., 2016) scales. Both these
studies used climate averaged over ~3.75° spatial resolution. The CLASS-CTEM framework
consists of the Canadian Land Surface Scheme (CLASS) coupled to the Canadian Terrestrial
Ecosystem Model (CTEM) which is a dynamic vegetation model.

In this paper, we evaluate the competition module of the CLASS-CTEM modelling framework at
the regional scale over the North American domain at 1° spatial resolution. This  resolution is
much finer than the 3.75° resolution used in the Melton and Arora (2016) study and therefore in



principle should allow a more realistic simulation of geographical distribution of PFTs as climate
niches are resolved.

The rest of this paper is organized as follows: Section 2 describes the CLASS-CTEM modelling
framework, details of the observation-based data and the experimental setup. Results are
presented in section 3 and a discussion follows in section 4. Finally, a summary and conclusions
are provided in section 5.

**2   Model, data and methods**

**2.1   CLASS-CTEM model**

The results presented here are obtained by coupling version 2.0 of CTEM (Melton and Arora,
2016), which dynamically simulates fractional coverage of its PFTs, to version 3.6 of CLASS
(Verseghy et al., 1993). CTEM simulates terrestrial processes for seven non-crop and two crop
PFTs (Table 1) and prognostically tracks carbon in three living vegetation components (leaves,
stems and roots) and two dead carbon pools (litter and soil). The terrestrial ecosystem processes
simulated in this study include photosynthesis, autotrophic respiration, heterotrophic respiration,
dynamic leaf phenology, allocation of carbon from leaves to stem and root components, fire, and
competition between PFTs which dynamically determines the fractional coverage of each PFT.
The amount of carbon in the leaf, stem and root components is used to estimate structural
attributes of vegetation. LAI is calculated from leaf biomass using PFT-dependent specific leaf
area (SLA) which determines area of leaves that can be constructed per kg C of leaf biomass
(Arora and Boer, 2005); vegetation height is calculated based on stem biomass for tree PFTs and
LAI for grass PFTs; and rooting depth is calculated based on root biomass (Arora and Boer,
2003). CTEM operates at a time step of one day except for photosynthesis and leaf respiration
which are calculated every 30 minutes for consistency with CLASS' energy and water balance
calculations which require stomatal resistance calculated by the photosynthesis module of
CTEM.





CLASS simulates the energy and water balance components at the land surface and operates at a
30 minutes time step. Liquid and frozen soil moisture and soil temperature are evaluated for
three soil layers (with maximum thicknesses of 0.1, 0.25 and 3.75 m). The actual thicknesses of
these permeable soil layers are determined by the depth to bedrock, which is specified on the
basis of the global data set of Zobler (1986). CLASS distinguishes four PFTs (needleleaf trees,
broadleaf trees, crops and grasses) which map directly to the nine PFTs represented in CTEM as
shown in Table 1. Needleleaf trees in CTEM are divided into deciduous and evergreen types,
broadleaf trees are divided into cold and drought deciduous and evergreen types, and crops and
grasses are divided into $C_3$ and $C_4$ types based on their photosynthetic pathways. In coupled
mode, CLASS uses the dynamically simulated vegetation attributes (including LAI, vegetation
height, canopy mass and rooting depth) and stomatal resistance calculated by CTEM, and CTEM
uses the soil moisture, soil temperature and net shortwave radiation calculated by CLASS. The
coupling frequency between CLASS and CTEM is one day.

2.1.1   **Competition parameterization**

Competition between PFTs in CTEM is parameterized following Arora and Boer (2006) who
presented a modified version of the LV equations. The approach is described in detail by Melton
and Arora (2016) and briefly summarized here. Consider, for simplicity, two PFTs that exist in a
grid cell with fractional coverages $f_1$ and $f_2$. Let PFT 1 represent a tree PFT and PFT 2 represent
a grass PFT. The bare fraction of grid cell not covered by any vegetation is represented by $f_B$. As
a result, $f_1 + f_2 + f_B = 1$. The rate of change of fractional coverages of the two PFTs and bare
fraction, for this example, are given by,

$$\frac{df_1}{dt} = c_1 f_1^{\beta}(1 - f_1) - m_1 f_1 \qquad (1)$$


$$\frac{df_2}{dt} = c_2 f_2^{\beta}(1 - f_1 - f_2) - c_1 f_1^{\beta} f_2 - m_2 f_2 \qquad (2)$$


$$\frac{df_B}{dt} = -c_1 f_1^{\beta} f_B - c_2 f_2^{\beta} f_B + m_1 f_1 + m_2 f_2 \qquad (3)$$






where $c_1$, $c_2$ and $m_1$, $m_2$ are the colonization and mortality rates for PFT 1 and PFT 2,
respectively. Colonization and mortality rates cannot be negative. Equations (1) and (2) show
that PFT 1 can invade the fraction covered by PFT 2 and the bare fraction; and that PFT 2 can
only invade the bare fraction. PFT 2 is not allowed to invade the fraction covered by PFT 1
because it is ranked lower than PFT 1. In CTEM, the superiority or ranking of the seven natural
non-crop PFTs is based on the tree-grass distinction and their colonization rates. Trees are always
considered to be superior than grasses because of their ability to shade them (Siemann and
Rogers, 2003). Within the tree and grass PFTs the dominance is determined dynamically based
on the colonization rate. The exponent β ($0 \leq β \leq 1$), an empirical parameter, controls the
behaviour of the LV equations. For β = 1, the equations represent the classical form of the LV
equations. The equilibrium fractional coverages for PFT 1 and 2 and bare fraction for this
classical form of the LV equations, denoted by $\tilde{f}_1, \tilde{f}_2$ and  are given by,

$$\tilde{f}_1 = \max\left\{\left(\frac{c_1 - m_1}{c_1}\right), 0\right\} \qquad (4)$$

$$\tilde{f}_2 = \max\left\{\left(\frac{(c_2 - m_2) - \left(1 + \frac{c_2}{c_1}\right)(c_1 - m_1)}{c_2}\right), 0\right\} \qquad (5)$$

$$\tilde{f}_B = \frac{(m_1 \tilde{f}_1 + m_2 \tilde{f}_2)}{(c_1 \tilde{f}_1 + c_2 \tilde{f}_2)} \qquad (6)$$

In equations (1) and (2), if the fractional coverages of PFT 1 and PFT 2 are initially zero then the
PFTs cannot expand for β = 1, implying that a minimum seeding fraction is always required.
Furthermore, in equation (5) as long as $(c_1 - m_1)$ is greater than $(c_2 - m_2)$ then the equilibrium
solution for $f_2$ will always be zero and PFT 2 will not be able to coexist with PFT 1. These
features of the classical form of the LV equations are avoided when $β = 0$, following Arora and
Boer (2006). The equilibrium fractional coverages for PFT 1 and 2 and bare fraction for the case
with $β = 0$ are given by,

$$\tilde{f}_1 = \left(\frac{c_1}{c_1 + m_1}\right) \qquad (7)$$

$$\tilde{f}_2 = \frac{c_2(1 - \tilde{f}_1)}{(c_1 + c_2 + m_2)} = \left(\frac{c_2 m_1}{(c_1 + m_1)(c_1 + c_2 + m_2)}\right) \qquad (8)$$





$$\tilde{f}_B = \frac{(m_1\tilde{f}_1 + m_2\tilde{f}_2)}{(c_1 + c_2)} \tag{9}$$


Unlike the classical version of the LV equations, the modified version of the equations with $\beta = 0$
does not require a minimum seeding fraction, and PFTs are able to increase their areal extent as
long as the climate is favorable and $c_i$ is positive. Also, as long as $m_1 > 0$ and $c_2 > 0$ then PFT 2
is able to coexist at equilibrium with PFT 1. Other values of $\beta$ between 0 and 1 give the dominant
PFT varying levels of access to sub-dominant PFTs but coexistence is most possible in the case
with $\beta = 0$.

The calculations of colonization and mortality rates are described in detail in Melton and Arora
(2016). Briefly, the colonization rate depends on the net primary productivity of a PFT. The
better a PFT performs for given climatic and soil conditions; the higher is its colonization rate.
The mortality rate represents the combined effect of four different processes: intrinsic or age-
related mortality, growth or stress mortality, mortality due to disturbance, and mortality due to
adverse climate which ensures that tree PFTs do not venture outside their bioclimatic zones.
**2.2 Forcing data**

The Climate Research Unit − National Centre for Environmental Prediction (CRU-NCEP)
reanalysis dataset (Viovy, 2012), is used to drive the model. The meteorological variables
(surface temperature, pressure, precipitation, wind, specific humidity, and incident short-wave
and long-wave radiation fluxes) are available at a spatial resolution of $0.5° \times 0.5°$ and at a six
hourly time interval for the period 1901-2010. These data are interpolated to 1° resolution
spatially, and disaggregated to half-hourly time resolution, a standard CLASS-CTEM model
integration time step. Temperature, pressure, wind, specific humidity, and long-wave radiation
are linearly interpolated in time while short-wave radiation is assumed to change with the solar
zenith angle with maximum radiation occurring at solar noon. Following Arora (1997) , the six-
hourly precipitation amount ($P$, mm/6-hour) is used to estimate the number of wet half-hours
($w_h$) in a given six-hour period for $P > 0$ as

$$w_h = \text{integer}(\max[1, \min(12, 2.6\log(6.93\,P))]). \tag{10}$$






The total precipitation amount is then distributed randomly but conservatively over these wet
half-hours. For instance, if seven out of 12 half hours intervals are calculated to be wet using
equation (10) then seven random numbers varying between 0 and 1 are generated and the six-
hourly precipitation amount is divided into seven parts in proportion to their respective random
numbers

Figure 1 shows the spatial distribution of mean annual precipitation and surface temperature over
the North American domain considered in this study. Mean annual precipitation values range
from less than 200 mm in the arid south-west United States and the high Arctic to more than
1500 mm on the Pacific coast. Mean annual temperature varies from around 24° C near the
southern limit of the domain in Mexico to less than -20° C in the Arctic tundra.

**2.3    Observation-based data**
2.3.1    **Fractional coverage of PFTs**

Observation-based estimates of fractional coverages of PFTs are based on a modified version of
the Wang et al. (2006) data set (hereafter WANG06) and the Moderate Resolution Imaging
Spectroradiometer land cover product (Friedl et al., 2013) (hereafter MODIS). These data are
used to evaluate the model results.

The WANG06 data set was developed for use by CTEM in simulations in which competition is
turned off and prescribed fractional coverage of PFTs is used. It combines observation- and
model-based data to estimate the annual change in fractional coverage of CTEM's nine PFTs
from 1850 to 2000. The Global Land Cover for the year 2000 (GLC2000), which is considered
as a base year for environmental assessment, divides the global land cover in 22 types is
available at 1 km resolution. WANG06 (their Table 2) mapped the GLC2000 data to CTEM's
nine PFTs aggregated to 0.5° resolution. The GLC2000 data were then extrapolated back to 1850
by adjusting the changes in crop area based on the then available Ramankutty and Foley (1999)
rop data set. Here, we use a modified version of the WANG06 data set which is based on the



HYDE v.3.1 crop data set (Hurtt et al., 2011) and generate an estimate of fractional coverage of
CTEM PFTs for the period 1850-2012.

The MODIS data set is based on the International Geosphere-Biosphere Programme (IGBP)
global vegetation data and University of Maryland's Science Data Set classification schemes at
0.25° spatial resolution. The data are derived from NASA HDF-EOS MODIS/Terra land cover
type. The data set is for the period 2001 to 2014 and contains 17 land cover types which we map
to CTEM's nine PFTs following the logic used in Wang et al. (2006) as shown in Table 2. The
fractional coverage of each of the nine CTEM PFT is first obtained at 0.25 degree resolution for
each year using the mapping scheme described in Table 2. These fractional coverages are then
re-gridded to the 1° spatial resolution for individual years. Finally, the data are averaged over the
period 2001-2014 to evaluate model results. MODIS data are known to exhibit substantial
interannual variability. Broxton et al. (2014), for instance, report that globally 40% of land pixels
show land cover change one or more times during 2001–2010 period. This does not necessarily
indicate changes in land cover but rather these differences are due to low accuracy in
categorizing the remotely sensed vegetation into one of the 17 MODIS land cover types, as
Broxton et al. (2014) note. This low accuracy is itself attributed to the fact that many landscapes
include mixtures of vegetation classes. Our re-gridding of fractional coverages to 1° spatial
resolution and averaging over the 2001-2014 time period to obtain climatology of land cover
alleviates some of the uncertainty since the effect of inaccurately classified land cover categories
is reduced due to both spatial and temporal averaging.

The separation of the broadleaf deciduous PFT into its drought and cold deciduous components
is performed via the approach used by WANG06. They assumed that below 24 °N deciduousness
is caused by soil moisture limitation and hence all broadleaf deciduous trees below this latitude
are drought deciduous, and above 34 °N deciduousness is caused by low temperatures and so all
broadleaf deciduous trees above this latitude are cold deciduous. Between 24 °N and 34 °N,
following WANG06 we assume a linear transition from drought deciduous to cold deciduous
trees. Finally, the separation of grasses into their $C_3$ and $C_4$ components is based on the
geographical distributions of the $C_3$ and $C_4$ fractions in the WANG06 data set.



### 2.3.2 **Gross primary productivity and LAI**

Observation-based estimates of gross primary productivity (GPP) are based on Beer et al. (2010). These data are based on the ecosystem level GPP obtained using eddy covariance measurements from more than 250 stations across the globe. Beer et al. (2010) extrapolated these eddy covariance flux data of GPP to the global scale using diagnostic models for the period 1982 – 2008, and the average over this time period is used to evaluate the model results. LAI data used for validation are the same as those used by Anav et al. (2013) and are based on Zhu et al. (2013) who use normalized difference vegetation index (NDVI) data from the Advanced Very High Resolution Radiometer (AVHRR) satellite to calculate average LAI for the period 1981 – 2010.

## 2.4 **Experimental setup**

### 2.4.1 **Equilibrium pre-industrial simulation**

The equilibrium pre-industrial simulation was initialized from zero biomass and zero fractional coverage for all non-crop PFTs. The fractions of $C_3$ and $C_4$ crop PFTs in each grid cell are specified corresponding to year 1850 based on the HYDE 3.1 dataset. The model was then run for 600 years driven by 1901-1925 CRU-NCEP climate data cycled repeatedly. These data do not show any warming trend (Wen et al., 2011) as opposed to the later part of the 20th century. Atmospheric $CO_2$ concentration was set to 285 ppm corresponding to the pre-industrial 1850 level. This pre-industrial equilibrium simulation yields initial conditions including fractional coverages of PFTs and carbon in all the live and dead pools for the transient 1850-2010 simulation. The 600 years simulation is sufficient for fractional vegetation cover and carbon pools to reach equilibrium.

### 2.4.2 **Transient historical simulation**

The transient historical simulation is performed for the period 1851-2010 and its carbon pools and fractional coverage of non-crop PFTs are initialized from the equilibrium pre-industrial simulation as mentioned above. The years 1851 to 1900 of this historical simulation are driven with CRU-NCEP climate data corresponding to the period 1901-1925, cycled twice. For the





period 1901-2010 the climate data corresponding to each year are used. Time varying
concentrations of atmospheric $CO_2$ are supplied for the period 1851-2010 based on the values
used in the fifth Coupled Modelling Intercomparison Project (CMIP5,
http://tntcat.iiasa.ac.at/RcpDb/) which are extended past 2005 to 2010 based on data from the
National Oceanic and Atmospheric Administration
(ftp://aftp.cmdl.noaa.gov/products/trends/co2/co2_annmean_gl.txt). The annual time-varying
fractional coverages of $C_3$ and $C_4$ crop PFTs in each grid cell are based on the HYDE 3.1 dataset.
The crop fractions in a grid cell are not available for colonization and neither are they subject to
disturbance by fire. Competition between PFTs occurs over the remaining non-crop fraction of a
grid cell. As total crop fraction in a grid cell changes over time (based on the HYDE 3.1 dataset)
the fractional area available for competition also changes.

The simulated results are evaluated against their observation-based counterparts using averaged
values over the last 30 years of the simulation corresponding to the period 1981-2010. This is the
same and/or very close to the time period for modified WANG06 land cover data set (1981-
2010), Beer et al. (2010) GPP (1982-2008), and Zhu et al. (2013) LAI (1981-2010). The only
exception is the MODIS-based land cover data which are available for the 2001-2014 period.
**3 Results**

**3.1 Continental scale values of PFT coverage**

Figures 2a compares the simulated vegetation areas summed over our North American domain
with the WANG06 and MODIS observation-based estimates. In the absence of another measure
of uncertainty, we use the range between these two observation-based estimates and assess if
simulated areal coverage of a given land cover type lies within or outside this range. The
simulated total vegetated area over North America ($14.8 \times 10^6$ km$^2$) is very similar to the
modified WANG06 ($14.4 \times 10^6$ km$^2$) and MODIS derived ($14.2 \times 10^6$ km$^2$). At the most basic
tree-grass-bare ground level, the simulated areas are closer to the MODIS-based estimates, than
to the estimate based on the modified WANG06 data. The simulated area covered by tree PFTs





$(7.8 \times 10^6 \text{ km}^2)$ is 6% lower than the MODIS derived estimate $(8.2 \times 10^6 \text{ km}^2)$ and 21% lower
than WANG06 $(9.7 \times 10^6 \text{ km}^2)$. The simulated grass coverage $(4.7 \times 10^6 \text{ km}^2)$ is 35% higher
than the MODIS derived estimate $(3.5 \times 10^6 \text{ km}^2)$. Both simulated and MODIS-based estimates
of area covered by grass PFTs are, however, substantially higher than the WANG06 $(2.4 \times 10^6$
$\text{km}^2)$ estimate. Averaged over the North American region, the simulated partitioning of land area
(excluding cropland area) covered by trees, grasses and bare ground (45%, 27%, 28%) is much
closer to the MODIS based data (48%, 20% and 32%) than to the modified WANG06 based data

345     (56%, 14%, 30%).


Figure 2b shows a comparison of simulated areas of individual PFTs with observation-based
estimates. This is a more stringent test of the performance of the competition module of CTEM.
The observation-based estimates of areas of all individual PFTs are available for the modified
WANG06 dataset. The MODIS based estimates were derived based on the mapping of MODIS'
17 land cover types to CTEM PFTs as shown in Table 2, which itself is mostly based on
WANG06. In Figure 2b, the observation-based estimates show that needleleaf evergreen (NDL
EVG) and broadleaf cold deciduous (BDL DCD CLD) are the dominant tree PFTs across North
America which is also shown by the model. The simulated total area of the NDL EVG tree PFT
$(3.9 \times 10^6 \text{ km}^2)$ is 28% less than WANG06 $(5.3 \times 10^6 \text{ km}^2)$ and 15% less than the MODIS based
estimate $(4.7 \times 10^6 \text{ km}^2)$. The simulated total area of BDL DCD CLD tree PFT $(3 \times 10^6 \text{ km}^2)$ is
13% less than WANG06 $(3.4 \times 10^6 \text{ km}^2)$ and 3% greater than MODIS based $(2.9 \times 10^6 \text{ km}^2)$
estimate. Overall, the model is able to capture the areas covered by individual PFTs reasonably
well. However, differences remain between observations-based and simulated estimates
especially the larger simulated area for $C_3$ grasses than both observation-based estimates.
Reasons for these differences include limitations in the model but also the manner in which
remotely-sensed vegetation is categorized into broad-scale vegetation types and then mapped
onto CTEM's nine PFTs, as discussed later.

In both Figures 2a and 2b although simulated areal coverages at the basic tree-grass-bare ground
level and for individual PFTs (except for $C_3$ grasses) are comparable to observation-based



estimates they are outside the range defined by difference of the WANG06 and MODIS based
estimates.

Figure 2c shows the time series of simulated areas summed over the domain covered by tree and
grass PFTs, the total vegetated area and the remaining bare ground. The specified area covered
by crop PFTs, based on the HYDE 3.1 data set, is also shown and first increases over the
historical period and then stabilizes and in fact somewhat decreases in association with cropland
abandonment over the north-eastern United States. The increase in the crop area results in a
decrease in the area covered by tree and grass PFTs up until the time when the crop area
stabilizes around 1970. In the model, this causes land use change emissions associated with
deforestation. After this time, as vegetation productivity responds to increasing atmospheric $CO_2$
concentration, the area covered by tree PFTs increases somewhat and colonizes available bare
areas and those covered by grass PFTs. This leads to a small reduction in the area covered by
grass PFTs as well as bare ground and the associated increase in the total vegetated area.

**3.2    Geographical distribution of PFTs**

3.2.1    **Total vegetated and bare ground fractions**

Figures 3 and 4 compare the geographical distribution of simulated total vegetated and bare
fractions across North America with the two observation-based estimates derived from the
modified WANG06 and MODIS data sets. The two observation-based estimates are also
compared amongst themselves. The metrics used are averaged root mean square difference
(RMSD) and spatial correlations ($R^2$).

The observation-based geographical distribution of vegetated fraction in Figure 3 (middle
column) shows densely vegetated land over the eastern part of the continent and less vegetation
coverage over colder regions in the North and drier regions in the south-central and south-west
United States. These broad scale patterns are consistent with the precipitation and temperature
climatologies of the region (Figure 1). The model reasonably reproduces the observed vegetation
distribution (left panel) with some obvious limitations. Simulated vegetation cover is

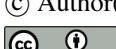



underestimated across the arid south-west United States, Great Plains and part of the Canadian
Prairies (right panel) due to lower simulated fractional coverage of tree and grass PFTs over
these regions (which are shown in the next section). The model overestimates vegetation
coverage in Northern Canada because of higher simulated grass cover in the Arctic as discussed
below in more detail. The spatial correlation and RMSD when comparing simulated vegetated
fraction to both observation-based estimates are 0.79 and around 18%, respectively. The spatial
correlation and RMSD between the two observation-based estimates themselves are 0.86 and
around 14%, respectively.

The simulated and observation-based bare ground fractions across North America are compared
in Figure 4. The observation-based estimates show that bare ground fraction is higher in Arctic
Canada and Alaska where, of course, cold temperatures limit vegetation growth and in the south-
west United States, Great Plains and the Prairies where low rainfall limits vegetation growth
(Figure 1). The biases in simulated bare ground fraction mirror those in the simulated vegetated
fraction but in an opposite manner. The model underestimates bare ground fraction across Arctic
Canada due to higher simulated grass cover as discussed in the next section. The model
overestimates the bare ground fraction generally across the arid and semi-arid south-west United
States, Great Plains and the Prairies. The spatial correlation and RMSD when comparing
simulated bare ground fraction to both observation-based estimates are around 0.46 and around
18%, respectively. The spatial correlation and RMSD between the two observation-based
estimates themselves are 0.68 and around 14%, respectively.

3.2.2 **Tree and grass cover**

Figure 5 compares the simulated tree cover with the two observation-based estimates. The model
reasonably reproduces the broad scale patterns including the Canadian boreal forest and the
temperate forests across the southeastern United States. However, the model simulates lower tree
cover across the western part of the continent compared to both observation-based estimates
particularly over the southwestern United States which is characterized by arid climate (Figure
1). The observation-based estimates do not particularly well agree over this region either. The
MODIS derived estimate suggests around 25% tree cover in the southwestern United States





while the WANG06 derived estimate suggests a tree cover of around 60% over a large area in
the region. The spatial correlation and RMSD when comparing simulated tree cover to both
observation-based estimates are around 0.68 and around 17%, respectively. The spatial
correlation and RMSD between the two observation-based estimates themselves are 0.75 and
around 15%, respectively. Possible reasons for differences between simulated and observation-
based estimates are discussed in detail in the discussion section and include the fact that the
CLASS-CTEM framework does not currently represent shrubs and there are limitations in the
observation-based data sets themselves. Shrubs are more prevalent in arid and semi-arid regions
where they are better suited to grow compared to both trees and grasses.

Figure 6 compares the geographical distribution of the simulated grass cover with the two
observation-based estimates. The broad geographical distribution of simulated grass cover
compares well with the two observation-based estimates with the notable exception of the Arctic
region including Alaska and northern Canada, where the model overestimates grass cover. This
overestimation of grass cover in the Arctic region is also the reason for the overestimation of
total vegetation fraction and the underestimation of bare fraction that was seen earlier in Figures
3 and 4 respectively.

As shown in Figure 6, the spatial correlation and RMSD when comparing simulated grass cover
to both observation-based estimates lie between 0.33 and 0.38 and between around 15-17%,
respectively. The spatial correlation and RMSD between the two observation-based estimates
themselves are 0.54 and around 9%, respectively. The two observation-based estimates disagree
most markedly over the western half of the United States where the MODIS derived estimates of
grass cover are higher.

3.2.3    **Needleleaf evergreen and broadleaf cold deciduous trees**

Figures 7a and 7b compare the geographical distribution of NDL EVG and BDL DCD CLD
trees, respectively, with their observation-based estimates. These two are the primary tree PFTs
which exist in the North American domain considered here.

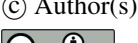



In Figure 7a, the overall simulated coverage of NDL EVG trees is lower than both observation-
based estimates as was also seen in Figure 2b. The simulated values are primarily lower in
western Canada and over a large area in the western United States according to estimates based
on the modified WANG06 data set. This is also the case along the wide swath of the Canadian
boreal forest. The model overestimates the coverage of NDL EVG trees in the eastern United
States. The spatial correlation and RMSD when comparing simulated coverage of NDL EVG
trees to both observation-based estimates lie between 0.36 and 0.40 and between around 16-17%,
respectively. The spatial correlation and RMSD between the two observation-based estimates
themselves are 0.52 and around 16%, respectively.

The geographical distribution of BDL DCD CLD trees is compared with its observation-based
estimates in Figure 7b. Although the simulated domain summed area of BDL DCD CLD trees (3
$\times$ $10^6$ km$^2$) is comparable to estimates based on the modified WANG06 (3.4 $\times$ $10^6$ km$^2$) and
MODIS (2.9 $\times$ $10^6$ km$^2$) data sets, there are two primary limitations in its simulated geographical
distribution. First, the simulated values are generally overestimated in Canadian boreal forests
and underestimated in the eastern United States. Second, the model simulates near zero coverage
in the arid south-western United States. The spatial correlation and RMSD when comparing
simulated coverage of BDL DCD CLD trees to both observation-based estimates are around 0.3
and around 12%, respectively. The spatial correlation and RMSD between the two observation-
based estimates themselves are 0.60 and around 8%, respectively.

### 3.2.4   C$_3$ and C$_4$ grasses

Figures 8a and 8b compare the simulated geographical distribution of C$_3$ and C$_4$ grasses with
observation-based estimates.

In Figure 8a, the most obvious limitation of the model is its excessive simulated grass coverage
in Alaska and in Arctic Canada. Other than this, the model reproduces the broad geographical
distribution of C$_3$ grasses including the Great Plains of United States and the Canadian Prairies,
where a large extent of grasslands is observed. The overestimated grass coverage at high
latitudes leads to a total   simulated C$_3$ grass area (4.4 $\times$ $10^6$ km$^2$) that is higher than estimates




based on the modified WANG06 ($1.9 \times 10^6$ km$^2$) and MODIS ($2.8 \times 10^6$ km$^2$) data sets. The
spatial correlation and RMSD when comparing simulated coverage of C$_3$ grasses to both
observation-based estimates lie between 0.34-0.38 and between around 15-17%, respectively.
The spatial correlation and RMSD between the two observation-based estimates themselves are
0.54 and around 12%, respectively.

Figure 8b shows the distribution of C$_4$ grasses which mostly occur in the tropics and do not
occupy large areas in North America (as was also seen in Figure 2b). The modelled geographical
distribution of C$_4$ grasses is larger than observation-based estimates but the absolute fractions
remain small so that the simulated area covered over the whole domain ($0.35 \times 10^6$ km$^2$) is
actually smaller than estimates based on the modified WANG06 ($0.45 \times 10^6$ km$^2$) and MODIS
($0.7 \times 10^6$ km$^2$) data sets. The spatial correlation and RMSD when comparing simulated
coverage of C$_4$ grasses to both observation-based estimates lie between 0.12-0.16 and between
around 3-5%, respectively. The spatial correlation and RMSD between the two observation-
based estimates themselves are 0.62 and around 5%, respectively.

### 3.2.5    Broadleaf evergreen and drought deciduous trees

The least prevalent PFTs in the North American domain considered here are broadleaf evergreen
(BDL EVG) and broadleaf drought deciduous (BDL DCD DRY) trees. As they are represented
in the model these are primarily tropical PFTs and hence generally do not exist above around 30
°N (see Figure 9), according to the bioclimatic limits used in the model for tree PFTs (Melton
and Arora, 2016). In our simulations, these PFTs therefore exist near the southern edge of the
United States. We do not evaluate spatial correlation and RMSD for these PFTs compared to the
two observation-based estimates for three reasons: 1) the geographical distribution of these PFTs
is limited to a small total area, 2) the geographical distribution of the BDL EVG tree PFT based
on observations cannot be directly compared to simulated values because, when mapping land
cover types to CTEM PFTs in WANG06, evergreen shrubs (which exist much farther north than
30 °N) are assigned to the the BDL EVG tree PFT, and 3) the geographical distribution of the
BDL DCD DRY tree PFT in the observation-based data sets is based on the arbitrary latitudinal
thresholds of 24 °N and 34 °N as mentioned earlier.






### 3.3 LAI and GPP


Figure 10 compares the geographical distribution of simulated LAI and GPP with observation-
based estimates for the present day.  In Figure 10a, the simulated geographical distribution of
LAI compares well with the observation-based estimates. The spatial correlation and RMSD
between simulated and observation-based estimates are 0.74 and 0.81 $m^2/m^2$, respectively. The
domain averaged simulated LAI of 2.5 $m^2/m^2$ is higher than the observation-based estimate of
2.1 $m^2/m^2$. The model captures the broad geographical patterns with higher LAI over the boreal
forest region in Canada and also in the eastern United States similar to observations. However,
some differences remain particularly over the drier southwest United States where the model
simulates bare ground with negligible LAI but observations suggest a small LAI of around 1
$m^2/m^2$. In contrast, the model slightly overestimates LAI over northern and Arctic Canada where
it simulates a higher fractional coverage of $C_3$ grasses, as seen earlier.

Consistent with the geographical distribution of LAI, the simulated GPP is overestimated in the
eastern United States and the Canadian boreal forest (Figure 10b). The broad geographical
distribution of GPP, similar to LAI, is consistent with the observation-based estimates. The
spatial correlation and RMSD between simulated and observation-based estimates are 0.78 and
225 $gC/m^2$.year, respectively. The domain averaged simulated GPP of 737 $gC/m^2$.year is higher
than the observation-based estimate of 628 $gC/m^2$.year. As with LAI, the simulated GPP is lower
than observations over the drier southwest region of the United States where the model simulates
more bare ground than observation-based estimates, and the model overestimates GPP over the
northern and Arctic Canada.

Figure 11 shows the time series of annual domain averaged GPP, LAI, net primary productivity
(NPP) and domain summed net biome productivity (NBP). The NBP term is essentially the net
atmosphere-land $CO_2$ flux which is the result of all terrestrial ecosystem processes including
photosynthesis, autotrophic and heterotrophic respiration, fire and land use change. NBP values
of zero indicate that the system is in equilibrium such that carbon gained by photosynthesis is
equal to carbon lost by respiration and other processes. Simulated GPP, LAI and NPP all show



an increase over the 20$^{th}$ century due to the increase in atmospheric $CO_2$ concentration and the
associated change in climate. The increase in $CO_2$ drives the increase in GPP and subsequently in
NPP and LAI through the $CO_2$ fertilization effect. The net result of this gradually increasing NPP
is that the terrestrial ecosystems become a sink of carbon and this is seen in the resulting positive
values of NBP. The simulated sink over the North American domain for the periods 1990-2000
and 2000-2010 is around 0.4 and 0.5 Pg C/year, respectively. Crevoisier et al. (2010) compare
the carbon sink over the North American region from five studies (their Table 1) for time periods
in the 1990s and 2000s. These reported sinks vary from 0.81±0.72 to 1.26±0.23 Pg C/year for the
period 1992-1996, 0.58 Pg C/yr for the period 2001-2006 and Crevoisier et al. (2010) themselves
estimate a value of 0.51 ± 0.41 Pg C/yr for the period 2004-2006. The sinks simulated by
CLASS-CTEM over the 1990s and 2000s are broadly consistent with these estimates.

**4 Discussion**

Allowing a terrestrial ecosystem model to simulate fractional coverages of its PFTs adds another
degree of freedom to the model compared to the case where the fractional coverages of its PFTs
are specified. This is a more stringent test of a model's performance. Errors in the simulated
geographical distribution of PFTs will, of course, lead to corresponding errors in the
geographical distribution of primary terrestrial ecosystem carbon pools and fluxes. Yet, the
CLASS-CTEM model is broadly able to reproduce the geographical distributions of GPP and
LAI. Limitations, of course, remain. In particular, the simulated LAI and GPP are high in Alaska
and in northern and Arctic Canada, and these variables are lower than their observation-based
estimates in arid regions of the western United States. The simulated fractional vegetation
coverage reflects these patterns.

It is difficult to conclusively determine whether these model limitations are due to the limitations
in the biogeochemistry parameterizations of the model for its existing PFTs or the simple
structural limitation that the model does not represent shrub, moss and lichen PFTs. Shrubs are
adapted to grow in arid and semi-arid regions, whether in cold or hot climates (where neither
grasses nor trees are able to grow) and their representation in the model would likely help to
increase the fractional vegetation cover in arid regions including those in the western United





States. At high latitudes grass growth is inhibited by mosses and lichens which flourish in cold
and damp conditions. A representation of moss and lichen PFTs and improved representation of
permafrost in the model would likely help to decrease simulated grass coverage in Arctic
regions. In the current version of the CLASS-CTEM model bioclimatic limits are used only for
tree PFTs to ensure that these PFTs do not venture outside their pre-determined bioclimatic
zones. In the model, bioclimatic limits are not used for grasses and their geographical
distribution is entirely the result of plant physiological processes and their competitive
interactions with the tree PFTs and amongst themselves. Since, in the Arctic region, grasses do
not face competition from tree PFTs, and moss and lichen PFTs are not represented in the model,
they are free to increase their expanse – climate permitting, of course. Another possible reason
for higher than observed grass coverage in the Arctic region is that in the current implementation
of CLASS only three permeable soil layers with maximum thicknesses of 0.1, 0.25 and 3.75 m
are represented and a boundary condition of zero heat flux is assumed across the bottommost
layer. This simple representation does not allow to model permafrost realistically. Permafrost is
more realistically modelled with multiple permeable and impermeable (extending into the bed
rock) layers that go sufficiently deep (> 30 m at least) to capture the slow evolution of soil
temperatures in response to climate warming (Teufel et al., 2017). The current set up of three
layers that go only 4.1 m deep produces soil temperatures that are warmer than in the set up
when permeable and impermeable layers are sufficiently deep and produces permafrost extent
that is lower than observation-based estimates (Koven et al., 2013). It is likely that warmly
biased soil temperatures in the current set up contribute to promote grass growth and allow it to
cover a larger area in the Arctic region than would be the case when permafrost is more
realistically modelled.

The lower than observed fractional vegetation cover in the arid and semi-arid regions of the
western United States, however, may not solely be due to model limitations alone. Here, we
argue that the manner in which remotely sensed land cover types are mapped to CTEM PFTs,
and the errors in calculating bare ground fraction in remotely sensed products also contribute to
mismatch between modelled and observation-based values of fractional vegetation cover. We
illustrate this by comparing the functional relationship between LAI and total vegetation cover.
Figure 12a shows this relationship for model simulated values. As expected, as LAI increases so





does the total vegetation cover. The relationship between these two variables is fairly tight in the
model and the green line is an exponential fit. The red dots in the figure correspond to grid cells
that lie in the region identified in the inset in Figure 12d and broadly correspond to the western
half of the United States. Figures 12b and 12c show the same relationship but between the
observation-based estimate of LAI from Zhu et al. (2013) (as mentioned in Section 2.3.2) and the
total vegetation cover based on the WANG06 and MODIS derived land cover data sets,
respectively. The blue and magenta lines in Figures 12b and 12c are the corresponding
exponential fits. When compared with Figure 12a, Figures 12b and 12c show much more scatter
around the fitted curves, and the overall relationship appears to break down for the red dots
corresponding to the grid cells in the western United States. A careful look at the red dots in
Figures 12b and 12c shows that the observation-based vegetation cover in the Western United
States for a large fraction of grid cells is around 60% regardless of the observation-based LAI
which ranges between 0.1 and 1.5 $m^2/m^2$. Clearly, it is physically unrealistic to achieve fractional
vegetation coverage of 60% below LAI values of 0.6 $m^2/m^2$ (the $m^2/m^2$ unit implies $m^2$ of leaf
area per $m^2$ of ground area) and this indicates that the fractional vegetation cover in this region is
likely overestimated in both observation-based data sets.

There are at least two ways in which errors in total vegetation cover can occur. The first relates
to the method by which the fractional vegetation cover is calculated for the land cover types in
the original remotely sensed land cover products: that is, for the 22 land cover types in the
GLC2000 data set upon which the WANG06 data are based and the 17 land cover types in the
MODIS data set. An example of such an error for arid regions is illustrated by Lawley et al.
(2014) who suggest that the MODIS soil fractional cover product, at least in its present form, is
unsuited to monitoring sparsely vegetated arid landscapes and generally unable to separate soil
from vegetation in situations where normalized difference vegetation index (NDVI) is low. The
second way in which errors are introduced is through the mapping of the remotely sensed land
cover types to the CTEM PFTs following Table 2 of WANG06 for the GLC2000 land cover
types, and following Table 2 in this manuscript for the MODIS land cover types.  This mapping
is based on available information in the literature but is also based on expert judgement which
introduces subjectiveness. For instance, it is debatable what fraction of the "open shrublands"
MODIS land cover type, which exists over much of the arid southwestern United States, is in



fact bare ground. In Table 2, we have allocated a value of 0.4 to this fraction following
WANG06. Had WANG06 allocated a higher value than this to bare ground, our simulated values
would have compared better with the observation-based values of bare ground fraction over arid
regions. Nevertheless this would not have changed the relationship, or rather the lack thereof,
between the observation-based estimates of LAI and the total vegetation cover in the western
half of the United States seen in Figures 12b and 12c.

The simulated areas covered by the primary two tree PFTs (NDL EVG and BDL DCD COLD)
have their weaknesses but large differences also exist between the two observation-based
estimates especially for the NDL EVG PFT. Modelling competition between two tree PFTs is
much more difficult than between trees and grasses. In the latter case trees are always considered
superior to grasses, but in the case of competition between two tree PFTs the superiority is based
on parameterized colonization rates which depend on simulated NPP. Based on comparisons
with observation-based estimates, the main limitation in model results here is that the model
overestimates the coverage of NDL EVG trees, and underestimates the coverage of BDL DCD
COLD trees in the eastern United States, while the opposite is true in western Canada. The
model, of course, does not represent individual species, while in the real world competition
occurs at the species level. One example of how additional PFTs in the CLASS-CTEM
framework can lead to improved model performance is illustrated by Peng et al., (2014). This
application of the model shows how sub-dividing the NDL EVG PFT into coastal and interior
types for the province of British Columbia in Canada leads to improvement in simulated LAI and
GPP. A recent attempt to explicitly represent physiological process in a model to simulate
competition between needleleaf and broadleaf cold deciduous trees at a regional scale is
illustrated in (Fisher et al., 2015) who incorporated the concepts from the Ecosystem
Demographics (ED) model into the community land model – dynamic vegetation model (CLM-
DGVM). Their results provide some interesting insights; however, validation of this approach at
the global scale over a wide range of PFTs remains challenging.

Finally, one of the objectives of this study was to evaluate if resolving climate niches by
performing CLASS-CTEM simulation at a finer resolution of 1° in this study allowed improved
simulation of geographical distribution of PFTs than in the Melton and Arora (2016) study that





evaluated the competition module of the CLASS-CTEM model at 3.75° spatial resolution at the
global scale. Figures 2 through 4 of Melton and Arora (2016) compare simulated geographical
distributions of PFTs with WANG06 data. Comparing their results over North America with
ones obtained here we note that the primary model limitations remain unchanged in the
application of the model at both spatial resolutions. These include lower simulated fractional
vegetation coverage in the arid south-west North American region and higher in the Arctic
region (due to higher grass coverage). In addition, in both applications of the model the
differences in simulated geographical distribution of NDL EVG and BDL DCD CLD PFTs,
compared to the WANG06 land cover data, are also similar. Model differences, compared to the
WANG06 data, therefore remain more or less similar in the application of the model at both
spatial resolutions.

The comparison between observation-based and simulated fractional coverages is the most
robust at the basic tree-grass-bare ground level. The subjectiveness introduced in the process of
mapping remotely sensed land cover types to the PFTs represented in a model, as mentioned
above, makes the comparison of simulated and observation-based fractional coverages for
individual PFTs less robust. Nevertheless, comparisons with observations allow useful insights
into model limitations as we have seen here.


## 5    Summary and conclusions

This study evaluates the CLASS-CTEM simulated fractional coverages of PFTs, when driven
with observed climate forcing, against the observation-based estimates from MODIS and the
modified WANG06 data sets over the North American region. In the past, performance of the
competition module of the CLASS-CTEM modelling framework has been assessed at global
scale, at a coarse spatial resolution of 3.75° (Melton and Arora, 2016), as well as at point scale,
for a range of locations across the globe (Shrestha et al., 2016). Our objective here was to assess
the performance of the CLASS-CTEM competition module at a higher spatial resolution of 1°
over North America. To achieve this objective we compared simulated present day geographical



distributions of fractional coverages of PFTs, but also LAI and GPP with their observation-based estimates.

The CLASS-CTEM modelling framework is generally able to reproduce the dominant features of the geographic distribution of PFT coverage, and LAI and GPP over the North American region. After 1960, the model simulates increasing GPP and LAI in response to changing climate as well as increased atmospheric $CO_2$ concentrations and the resulting sink for the 1990s and 2000s is broadly consistent with other estimates.

The simulated geographical distribution of PFTs, when compared to observation-based estimates, show two primary limitations which are excessive grass cover in the Arctic region and low vegetation cover in the arid western United States, although for the latter the observation-based estimates themselves may have their own weaknesses. There are three main factors in the CLASS-CTEM modelling framework that may have contributed to these differences: 1) the absence of a shrub PFT, which we believe is the reason for low simulated vegetation coverage in the arid to semi-arid western United States, 2) the absence of moss and lichen PFTs that may inhibit the establishment of grasses, and 3) probably a lack of sensitivity of $C_3$ grasses to high latitude climate and an inadequate representation of permafrost. Future model developments will focus on these aspects with a view to improving model performance.



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



Table 1: Plant functional types (PFTs) represented in CTEM and their relation to CLASS PFTs.


| CLASS PFTs | CTEM PFTs | CTEM PFT Symbol |
|---|---|---|
| Needleleaf trees | Needleleaf Evergreen trees | NDL-EVG |
| | Needleleaf Deciduous trees | NDL-DCD |
| Broadleaf trees | Broadleaf Evergreen trees | BDL-EVG |
| | Broadleaf Cold Deciduous trees | BDL-DCD-CLD |
| | Broadleaf Drought/Dry Deciduous trees | BDL-DCD-DRY |
| Crops | $C_3$ Crops | CROP-C3 |
| | $C_4$ Crops | CROP-C4 |
| Grasses | $C_3$ Grasses | GRASS-C3 |
| | $C_4$ Grasses | GRASS-C4 |



Table 2: Reclassification of the 17 MODIS land cover classes into the nine CTEM PFTs

| SN | Items | Tree | | | | Crop | Grass | Bare | Reference |
|---|---|---|---|---|---|---|---|---|---|
| | | NDL EVG | NDL DCD | BDL EVG | BDL DCD | | | | |
| 1 | Woody Savanna | | | 0.1 | 0.4 | | 0.25 | 0.25 | Dai et al. (2001) |
| 2 | Water bodies | | | | | | | 1 | |
| 3 | Urban built up areas | 0.05 | | | 0.05 | | 0.1 | 0.8 | Dai et al. (2001) |
| 4 | Savanna | | | 0.05 | 0.3 | | 0.4 | 0.25 | Wang et al. (2006) |
| 5 | Permanent Wetlands | | | | | | 0.25 | 0.75 | Dai et al. (2001) |
| 6 | Permanent snow and ice | | | | | | | 1 | Wang et al. (2006) |
| 7 | Open Shurblands | 0.1 | | | 0.15 | | 0.35 | 0.4 | Wang et al. (2006) |
| 8 | Needleleaf evergreen | 1 | | | | | | | Wang et al. (2006) |
| 9 | Needleleaf deciduous | | 0.8 | | | | 0.1 | 0.1 | Wang et al. (2006) |
| 10 | Mixed forest | 0.45 | | | 0.45 | | 0.1 | | Wang et al. (2006) |
| 11 | Grasslands | | | | | | 0.65 | 0.35 | Wang et al. (2006) |
| 12 | Croplands | | | | | 0.9 | | 0.1 | Wang et al. (2006) |
| 13 | Cropland natural veg. mosaic | | | 0.2 | | 0.5 | 0.2 | 0.1 | Wang et al. (2006) |
| 14 | Closed shrublands | 0.2 | 0.2 | | 0.4 | | 0.2 | | Wang et al. (2006) |
| 15 | Broadleaf evergreen | | | 1 | | | | | Wang et al. (2006) |
| 16 | Broadleaf deciduous | | | | 1 | | | | Wang et al. (2006) |
| 17 | Bare ground | | | | | | | 1 | Wang et al. (2006) |





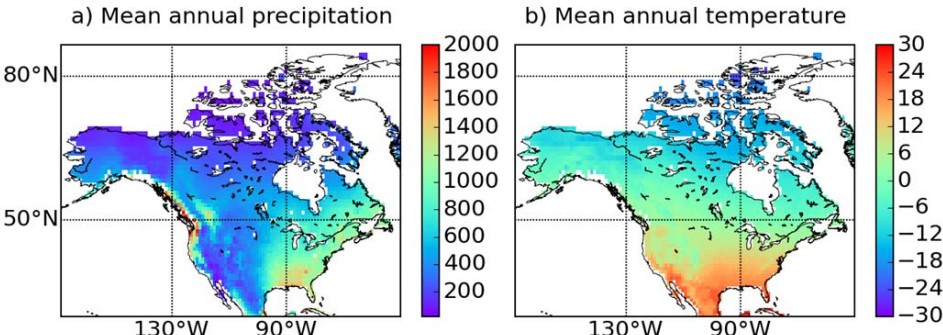

Figure 1. Spatial distribution of mean annual a) precipitation (mm), and b) temperature (°C)
across North America. Grid cells with permanent ice/glaciers have been masked out.





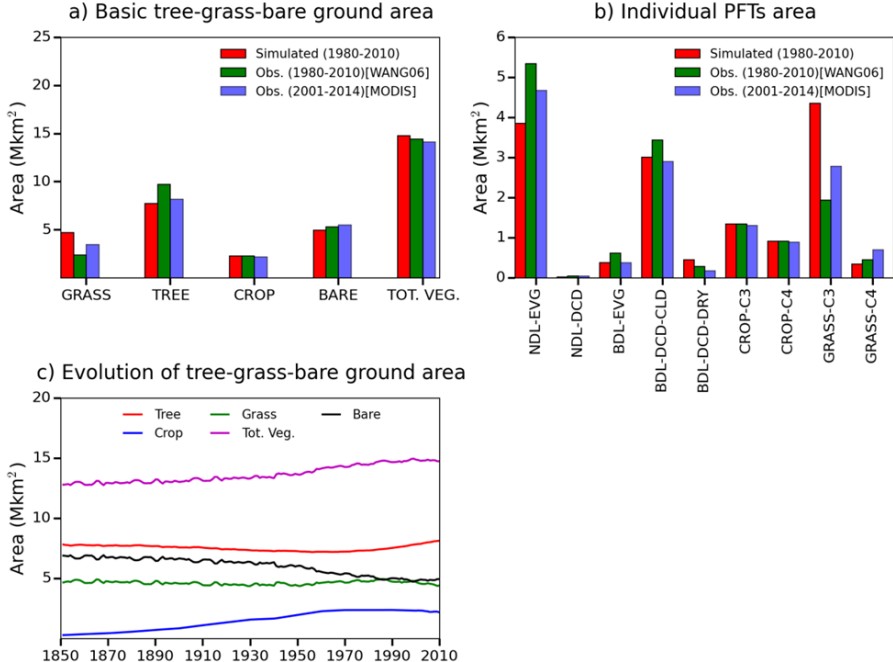

Figure 2. Comparison of observation-based and simulated vegetation areas summed over the
North American domain a) grass, treed, crop, bare ground and total vegetated area, b) individual
PFT areas, and c) evolution of simulated vegetation areas summed over the domain.





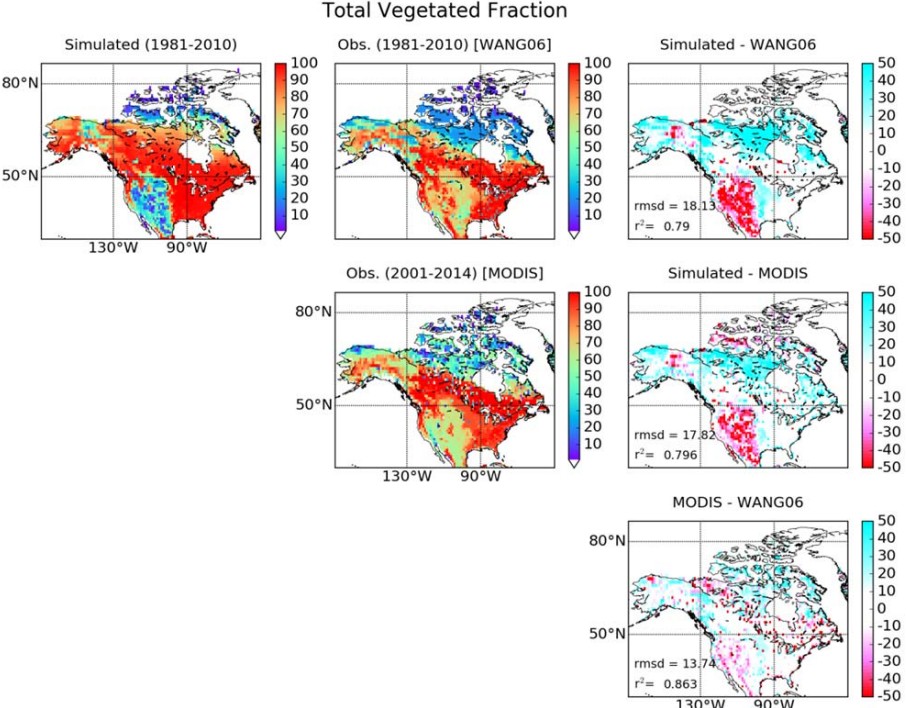

Figure 3. Spatial distribution of total vegetated coverage across North America. Simulated,
observation-based, and differences are presented in the left, middle and right columns,
respectively. The differences column includes model biases with respect to WANG06 (top panel)
and MODIS (middle panel), and the difference between the two observation-based estimates
(bottom panel). Root mean square difference (rmsd) and coefficient of determination ($r^2$) are also
shown in each case.





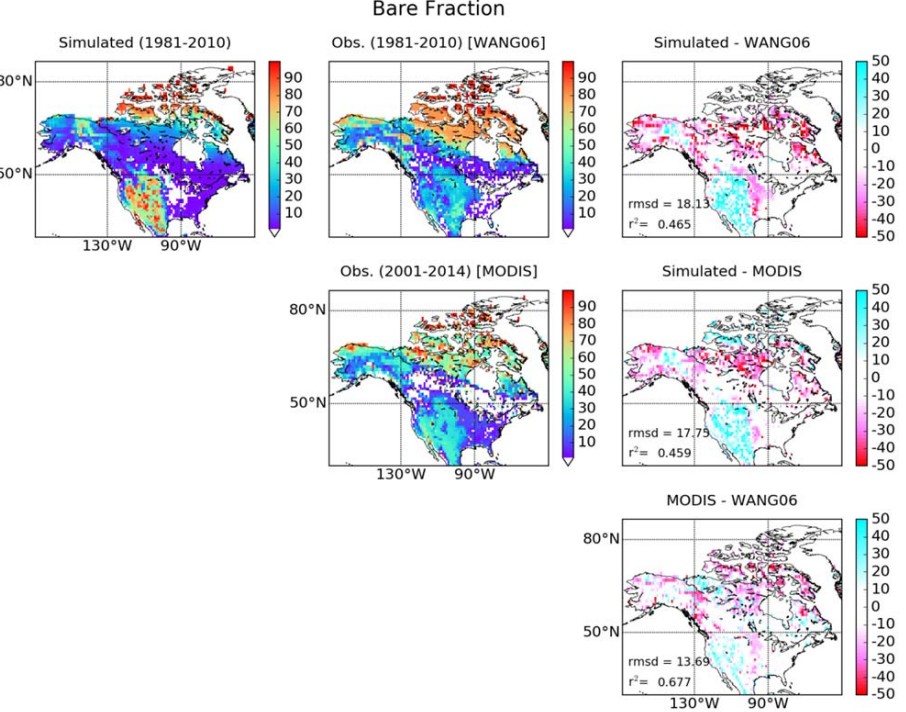

Figure 4. Spatial distribution of bare ground coverage across North America. Simulated,
observation-based, and differences are presented in the left, middle and right columns,
respectively. The differences column includes model biases with respect to WANG06 (top panel)
and MODIS (middle panel), and the difference between the two observation-based estimates
(bottom panel). Root mean square difference (rmsd) and coefficient of determination ($r^2$) are also
shown in each case.





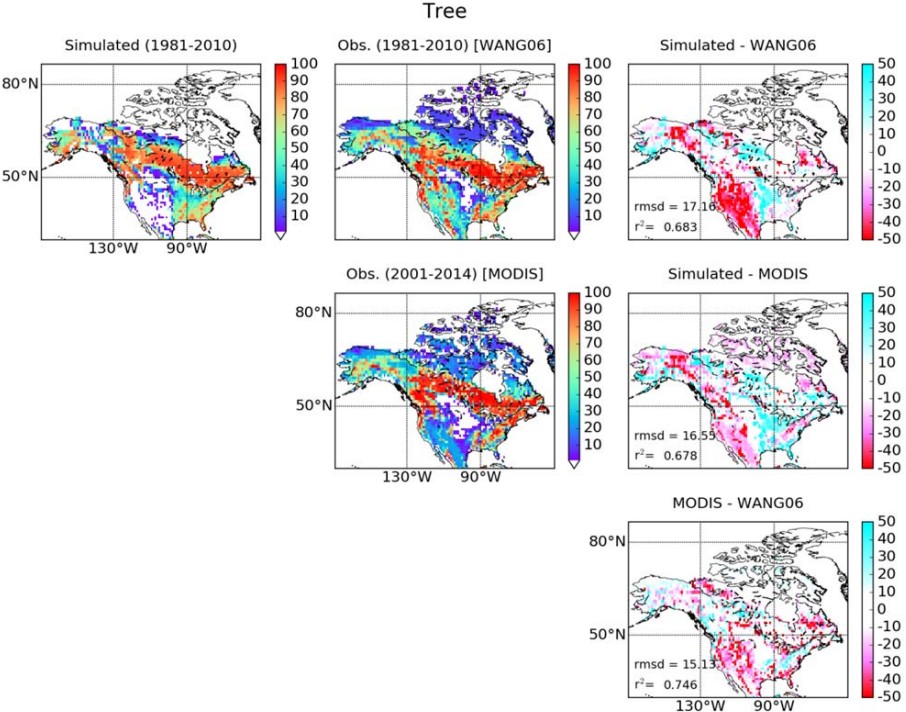

Figure 5. Spatial distribution of tree coverage across North America. Simulated, observation-based, and differences are presented in the left, middle and right columns, respectively. The differences column includes model biases with respect to WANG06 (top panel) and MODIS (middle panel), and the difference between the two observation-based estimates (bottom panel). Root mean square difference (rmsd) and coefficient of determination ($r^2$) are also shown in each case.





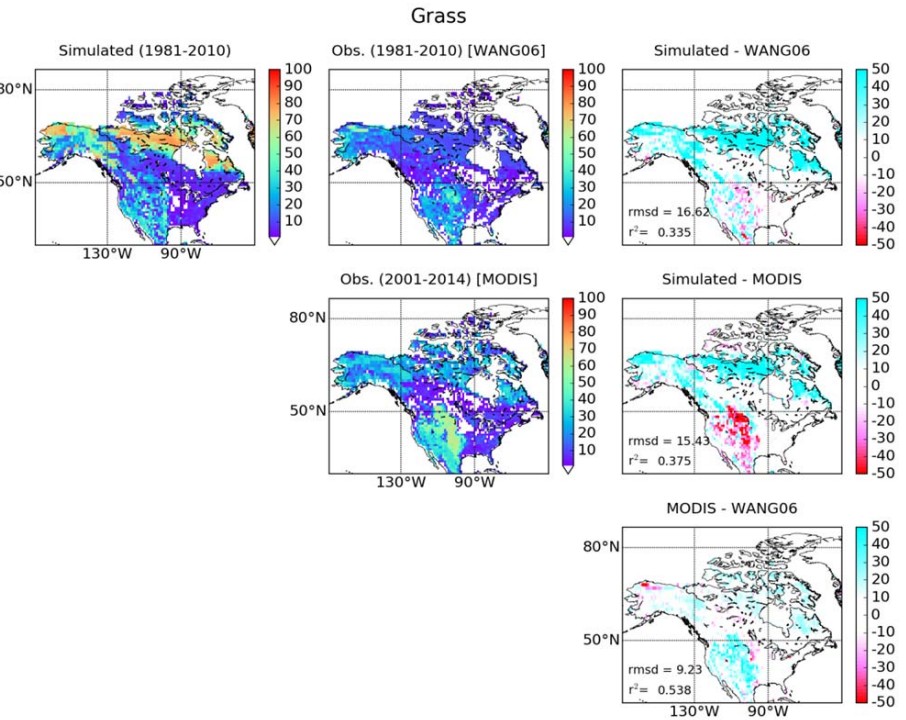

Figure 6. Spatial distribution of grass coverage across North America. Simulated, observation-based, and differences are presented in the left, middle and right columns, respectively. The differences column includes model biases with respect to WANG06 (top panel) and MODIS (middle panel), and the difference between the two observation-based estimates (bottom panel). Root mean square difference (rmsd) and coefficient of determination ($r^2$) are also shown in each case.








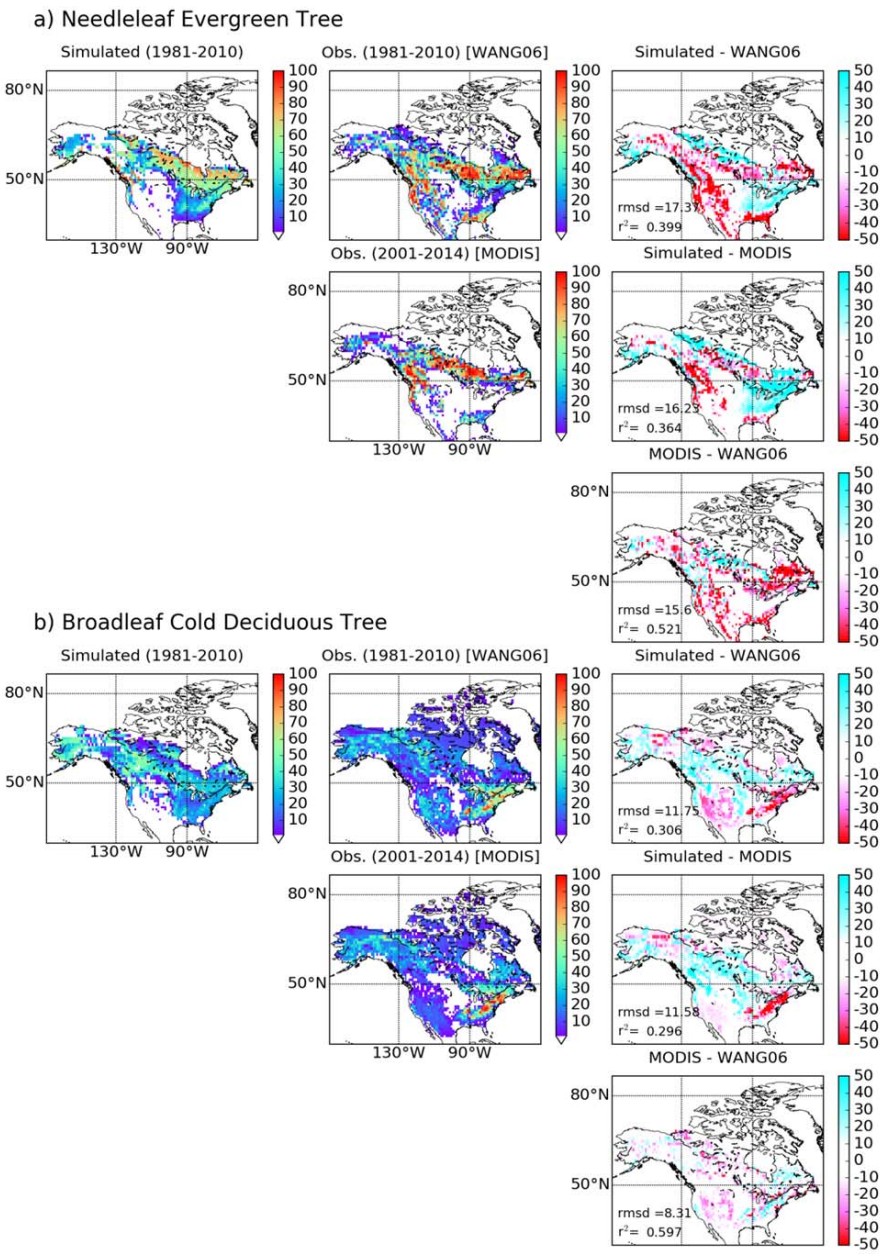


Figure 7. Spatial distribution of a) needleleaf evergreen tree, and b) broadleaf cold deciduous tree across North America. Simulated, observation-based, and differences are presented in the left, middle and right columns, respectively. The differences column includes model biases with respect to WANG06 (top panel) and MODIS (middle panel), and the difference between the observation-based estimates (bottom panel). Root mean square difference (rmsd) and coefficient of determination ($r^2$) are also shown in each case.





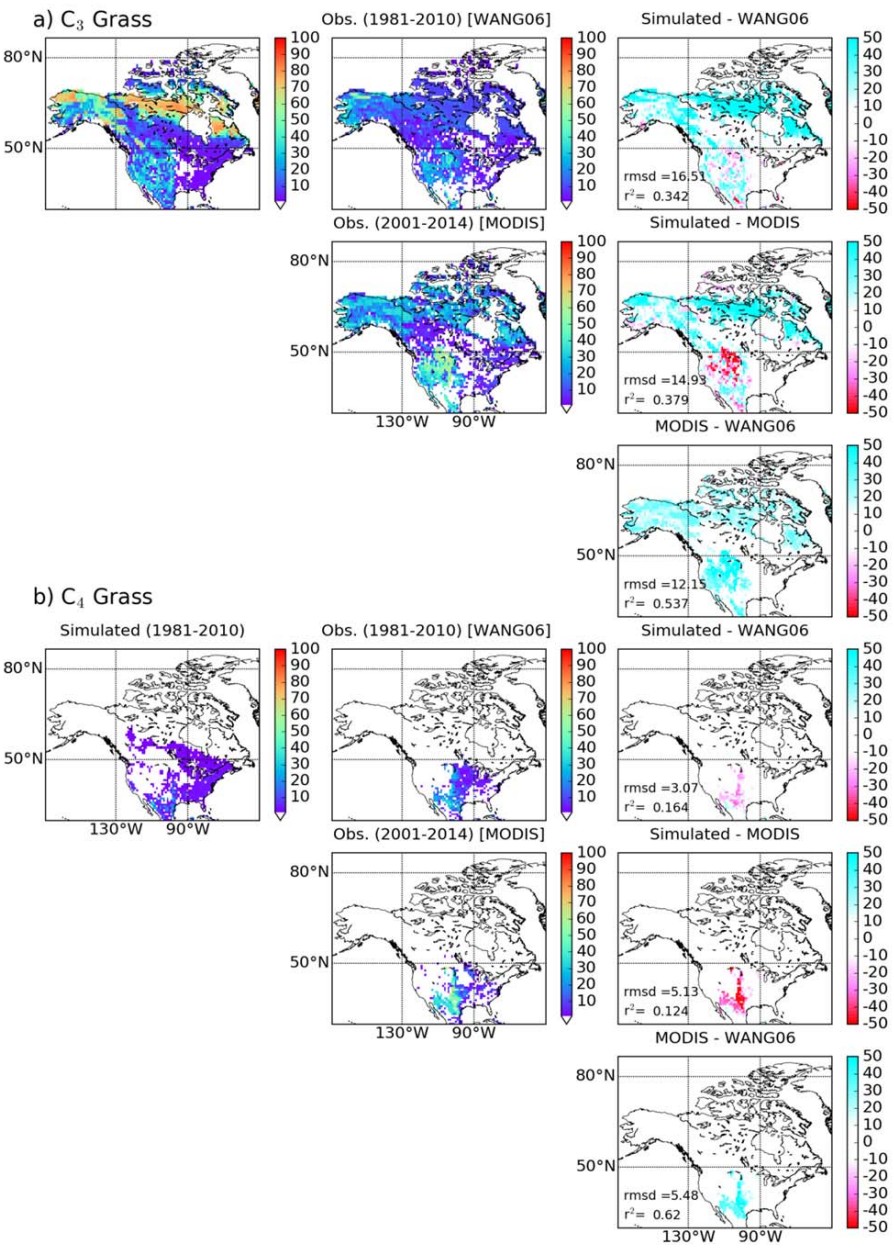

59

Figure 8. Spatial distribution of a) $C_3$ grasses, and b) $C_4$ grasses across North America.
Simulated, observation-based, and differences are presented in the left, middle and right
columns, respectively. The differences column includes model biases with respect to WANG06
(top panel) and MODIS (middle panel), and the difference between the the observation-based
estimates (bottom panel). Root mean square difference (rmsd) and coefficient of determination
($r^2$) are also shown in each case.




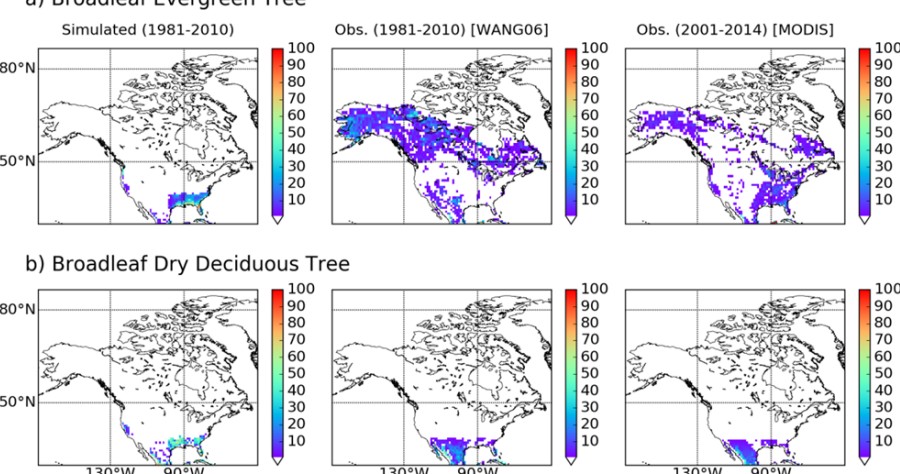



Figure 9. Spatial distribution of a) broadleaf evergreen tree, and b) broadleaf dry deciduous tree
across North America. Simulated, WANG06 and MODIS distribution are presented in the left,
middle and right columns, respectively.










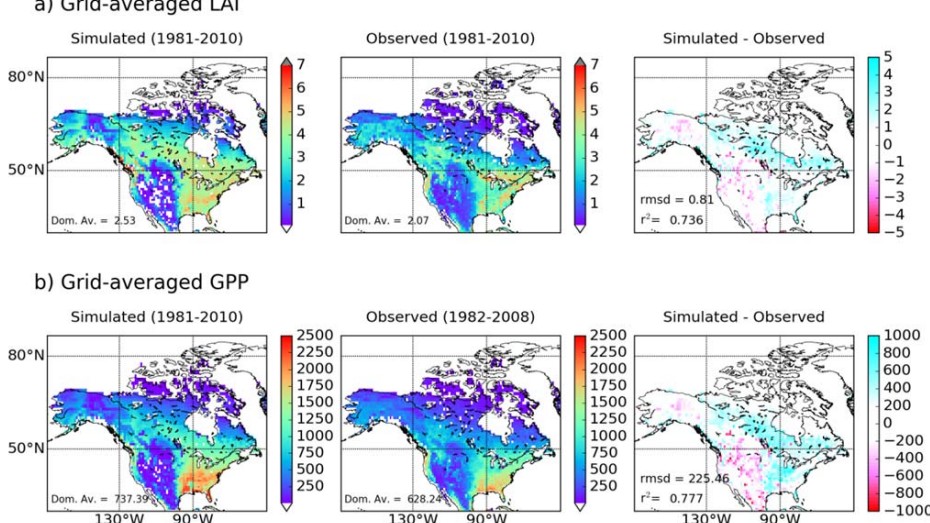



Figure 10. Spatial distribution of a) grid averaged maximum LAI ($m^2$ $m^{-2}$), and b) grid averaged GPP (g C $m^2$ $y^{-1}$) across North America. Simulated, observation-based, and differences between them are presented in the left, middle and right columns, respectively. Root mean square difference (rmsd) and coefficient of determination ($r^2$) are also shown in each case.





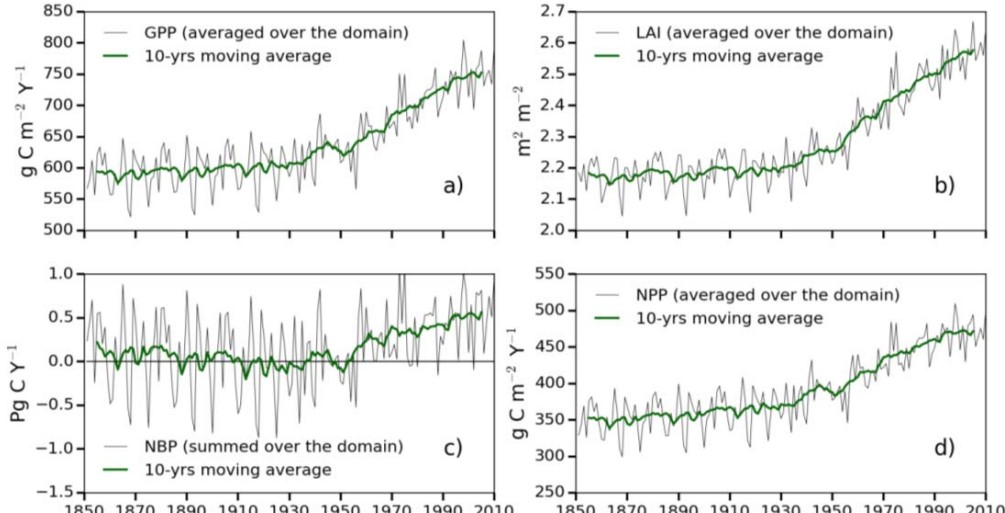


Figure 11. Time series evolution of a) domain averaged GPP (g C m$^{-2}$ y$^{-1}$), b) domain averaged

LAI (m$^{2}$ m$^{-2}$), c) domain total NBP (Pg C m$^{-2}$ y$^{-1}$), and d) domain averaged NPP (g C m$^{-2}$ y$^{-1}$).






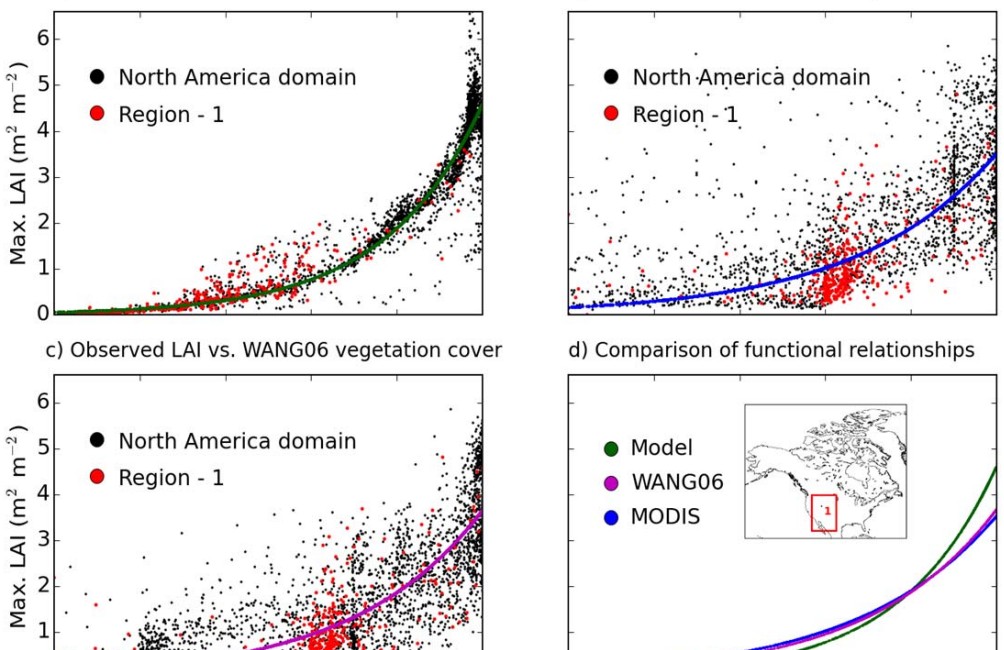



Figure 12. Scatter plots of a) simulated LAI vs. simulated total vegetation coverage, b) observed LAI vs. MODIS-derived total vegetation coverage, c) observed LAI vs. WANG06 total vegetation coverage. Plot d) shows a comparison of the fitted curves represented by solid lines, with an inset map of North America showing the sub-domain of interest bounded by a red rectangle.

104

105