# Peer review of "An assessment of geographical distribution of different plant functional types"

_Biogeosciences, 2017_

## Referee Comment (RC1) · Anonymous Referee #1 · 17 Apr 2017

General comments.

R. K. Shrestha and coauthors report the results of testing the CTEM model ability to simulate plant functional types (PFT) distribution over North America at the spatial resolution of 1 degree in the changing climate of 19th and 20th centuries. The results confirm the ability of the model to broadly reproduce the distribution of the major PFTs, LAI, and carbon fluxes. The paper is well written and is suitable for publication after minor revision.

Detailed comments.

[Figure]

1. The CTEM plant competition scheme is intended to simulate effects of climate change on vegetation composition. One aspect of balance between broadleaved deciduous and needle-leaved evergreen PFTs doesn't seem to be addressed here and in (Melton and Arora, 2016). As climate warms up in the boreal region, late succession broadleaved deciduous species are expected to get advantage over late succession needle-leaved evergreen trees. However some needle-leaved evergreen species like jack pine occupy ecological niche of nutrient-poor sandy soils, where change to late-succession broadleaves is less likely. Thus the approach of lumping together (see line 75 in text) "fir (Abies), spruce (Picea) and pine" carries some disadvantages. If there is proper data for the North America, it is useful to provide estimates of the forest area fraction, where current completion scheme may underestimate the resilience of needle-leaved evergreens.

2. Figure 5 shows tree coverage, and much of wetland-dominated Hudson Bay lowland appears covered with trees on both simulated and MODIS maps. Known weakness of the MODIS or GLC2000 vegetation maps is their inability to explicitly represent boreal peatlands. As a result, peatland area is assigned to mixture of grass and tree PFTs, and the model is forced to simulate competition between them as if they grow on mineral soil. In reality tree PFTs perform quite differently on peatland surface, where productivity and biomass are typically lower than in nearby well-drained landscapes. Alternatively to using MODIS or GLC2000 data, wetland area fraction dataset by Matthews and Fung (1987) or Global Lakes and Wetlands Database (Lehner and Doll 2004) are readily available and provide a better view of peatland area fraction than MODIS or GLC2000 vegetation products. As GLC2000 and GLWD have similar resolution, it is possible to separate wetland pixels from the rest. Authors mentioned lack of moss and lichen cover PFT in the model. However, fortunately for DGVMs, the peatland area changes slowly over millennial scale, and vegetation changes there can be considered separately from those occurring on mineral soil. Thus, for fair comparison it is better to omit peatland/wetland area when comparing the modeled vegetation distribution (tree vs grass PFTs) with observations, if that is technically possible.

3. Figure 9 shows broadleaved evergreen tree coverage. While simulated pattern looks reasonable, the observed one shows some unexpected broadleaved evergreen presence in the North of Canada and Alaska. Although authors point to the map of Wang et al 2006 for assigning the northern evergreen shrubs to the broadleaved evergreen tree PFT, it is recommended to correct the map, and remove broadleaved evergreen tree type north of its known area before comparing with model simulation.

4. In the introduction (line 96-97), authors define using 1 degree spatial resolution as major difference between this and Melton and Arora (2016) studies. Using higher spatial resolution appears as a major motivation for the study. However there is no clear statement or conclusion on the effect of resolution on improving simulated distributions of PFT, LAI or carbon fluxes. Even if there is no significant improvement, which is possible, the result should be stated clearly.

Minor, technical corrections

Line 35. Authors point that some processes need further development. Suggest to briefly mention which actual processes need attention.

References

Lehner B. and Döll P., Development and validation of a global database of lakes, reservoirs and wetlands, J. Hydrol., 296, 1–22, 2004

Matthews, E., and I. Fung, Methane emission from natural wetlands: Global distribution, area, and environmental characteristics of sources, Global Biogeochem. Cycles, 1(1), 61–86, doi:10.1029/GB001i001p00061, 1987.

---

## Referee Comment (RC2) · Anonymous Referee #2 · 5 Jun 2017

This paper evaluates the performance of the competition module of the CLASS-CTEM modeling framework at the fine spatial scale of 1 degree by comparing the geographical distribution of vegetation types as simulated by the model with observation data. The paper also proceeds to try and identify the main reasons why the simulated results deviate from the observed patterns. The research represented by this paper is very important to the scientific community as it (1) documents the geographical regions where the model does and does not perform well, (2) by seeking to establish the reasons for deviations from observation, sheds more light on inter-model (ESM) uncertainty

and (3) finds out the response of terrestrial vegetation to increasing atmospheric CO2 and a changing climate. However, the paper failed to evaluate the performance of the competition module of the model on certain key aspects and did not do thorough job of assessing the reasons for model deviation from observations. I would suggest a substantial revision of the paper.

My main/general comments are:

1. Establishing reasonability: The primary objective of a model is to 'look' at the terrestrial biosphere for such conditions that can not be observed. For example, experimental manipulations of ecosystems, simulations for climatic conditions in the future. It would be highly improbable to have model outputs exactly match observational data. Two independent observational datasets have differences. Therefore, while assessing model performance, it is essential to establish quantitatively what 'reasonable performance' is. It could be a measure of deviation from multi-observational mean (ensemble) or a threshold for spatial correlation (R2). Establishing this 'reasonability' will help users of this model to limit themselves to assessing simulation results for certain geographical regions and/or certain Plant Functional Types (PFTs).

2. More comprehensive model performance assessment: a) A critical element of model evaluation that is missing is comparison with other published DGVM outputs. DGVMs are similar in their basic structure but differ with respect to certain processes and/or parameterization. Thus comparison with other DGVM outputs will allow the authors to establish how the carbon cycle dynamics and/or terrestrial vegetation patterns of CLASS-CTEM compare with that of other models. This will further help to establish the 'reasonability' of CLASS-CTEM. I would suggest using the ensemble of DGVM outputs from the TRENDY project, which is a consortium of DGVM groups who set up a project to investigate further the spatial trends in Net Biome Production (NBP) and agreed to perform a factorial set of DGVM simulations over the historical period, 1901 – 2010 (see http://dgvm.ceh.ac.uk/ ). b) Since the primary objective is to compare model performance, it will be a good idea to compare the model GPP to a satellite based model

output like the MODIS MOD17 data (see http://www.ntsg.umt.edu/project/MOD17/ →
'Data Product'). Using this will also help in establishing observational uncertainty.

3. Regional analysis: The paper currently does perform a regional analysis, but it is qualitative. The future users of CLASS-CTEM would want to know in which specific regions of North America is the model performance reliable enough. I would thus suggest using shapefiles to demarcate the different regions being studied and then carry out statistical comparisons for each region. It would be helpful for the reader if the maps of individual regions are explicitly shown in the paper.

4. Time series comparison: According to the authors, the primary goal of the paper is to evaluate the performance of the competition module of CLASS-CTEM. While comparing current vegetation patterns and productivity with that of observational data is a good way to evaluate competition, it is not comprehensive since it looks at only one time slice and equilibrium conditions. Competition can be observed in reality when an ecosystem is subjected to environmental changes and/or disturbance. The time-series data of both CLASS-CTEM data and observational data both incorporate the response of the terrestrial ecosystems to changes in the climate and atmospheric $CO_2$ concentrations. Thus a better evaluation of the competition module should include equivalent time series comparisons of model and observation.

5. Attributing deviations from observation: The authors admit that it is difficult to determine whether the model limitations are due to biogeochemical parameterizations or structural limitations (Pg 20, lines 578-580). If this is the case, then attributing deviations from observations is mere speculation. While speculation backed by scientific logic is okay for the discussions section, it shouldn't be present in the abstract since this isn't something that the authors have assessed. That being said, model limitations can be diagnosed by tuning parameters. The biogeochemical parameter that is the main suspect causing deviations from observation should be identified and parallel simulations should be run using varying values of the parameter and/or using values which are a closer match to North American species.

6. Impact of change in spatial scale: The authors say that one of the goals of the paper was to look at how the 1 degree version of the model changed results when compared to the earlier study performed using the 3.75 degree version (Pg 23, lines 674-678). The authors make qualitative statements and not any quantitative assessments. I was expecting to see RMSD and R2 values for both model versions. This would help establish whether a higher spatial resolution helps to improve model performance or not.

Minor comment:

The paper would be of much more value to the climate science and/or carbon cycle community if the authors used more of carbon cycle flux and/or stock terms/values to assess CLASS-CTEM performance. While it is useful to have a detailed analysis of the major PFTs, analyzing the minor PFTs doesn't add much value to the model assessment. Instead grouping PFTs under broader categories (eg. Trees/grass and/or needleleaf/broadleaf trees) adds value to the paper as it makes comparisons with other models and datasets simpler.

---

## Author Comment (AC1) · 22 Jun 2017

We would like to thank the reviewers for their constructive comments. Our responses for each comment are as shown in bold below. Reviewers' comments are shown in normal text.

**Reviewer 1**

1) The CTEM plant competition scheme is intended to simulate effects of climate change on vegetation composition. One aspect of balance between broadleaved de-

ciduous and needle-leaved evergreen PFTs doesn't seem to be addressed here and in (Melton and Arora, 2016). As climate warms up in the boreal region, late succession broadleaved deciduous species are expected to get advantage over late succession needle-leaved evergreen trees. However some needle-leaved evergreen species like jack pine occupy ecological niche of nutrient-poor sandy soils, where change to late succession broadleaves is less likely. Thus the approach of lumping together (see line 75 in text) "fir (Abies), spruce (Picea) and pine" carries some disadvantages. If there is proper data for the North America, it is useful to provide estimates of the forest area fraction, where current completion scheme may underestimate the resilience of needle-leaved evergreens.

**The reviewer makes a valid point. In its current form the model is unable to capture the resilience of needle leaf evergreen species like Jack Pine which occupies ecological niches in nutrient-poor sandy soils and this aspect is not discussed in the manuscript. The two reasons for this are a lack of coupling of nitrogen (N) and carbon (C) cycles in the CLASS-CTEM modelling framework and the fact that competition is modelled at the plant functional type (PFT) level and not at the species level. Work is currently underway to include a representation of N cycle in CLASS-CTEM and at that point interactions between the C and N cycles will be able to modulate the competition between PFTs. However, representing competition between species for a model designed to be implemented at the global scale is a non-trivial exercise and not foreseeable in the near future. Tuning model parameters for hundreds of species at the global scale is currently not possible due to lack of available data at the species level nor there are remotely sensed land cover data available at the species level against which model simulated geographical extent of a species can be evaluated at the global scale. We will include additional discussion of these points in section 4 when revising our manuscript.**

2) Figure 5 shows tree coverage, and much of wetland-dominated Hudson Bay lowland

appears covered with trees on both simulated and MODIS maps. Known weakness of the MODIS or GLC2000 vegetation maps is their inability to explicitly represent boreal peatlands. As a result, peatland area is assigned to mixture of grass and tree PFTs, and the model is forced to simulate competition between them as if they grow on mineral soil. In reality tree PFTs perform quite differently on peatland surface, where productivity and biomass are typically lower than in nearby well-drained landscapes. Alternatively to using MODIS or GLC2000 data, wetland area fraction dataset by Matthews and Fung (1987) or Global Lakes and Wetlands Database (Lehner and Doll 2004) are readily available and provide a better view of peatland area fraction than MODIS or GLC2000 vegetation products. As GLC2000 and GLWD have similar resolution, it is possible to separate wetland pixels from the rest. Authors mentioned lack of moss and lichen cover PFT in the model. However, fortunately for DGVMs, the peatland area changes slowly over millennial scale, and vegetation changes there can be considered separately from those occurring on mineral soil. Thus, for fair comparison it is better to omit peatland/wetland area when comparing the modeled vegetation distribution (tree vs grass PFTs) with observations, if that is technically possible.

**This is also a valid point. It's a coincidence that both observation-based data show trees and grasses in the existing peatland areas along Hudson Bay and the model simulates the same since peatlands are not represented in our modelling framework yet. We will make a note of this in revising our manuscript although we do not feel that omitting the Hudson Bay area will change the correlation coefficients and RMSE significantly to change the conclusions drawn in our manuscript. Work is also under way to bring in a peatland model developed for CLASS-CTEM (Wu et al., 2016) into our modelling framework.**

3) Figure 9 shows broadleaved evergreen tree coverage. While simulated pattern looks reasonable, the observed one shows some unexpected broadleaved evergreen presence in the North of Canada and Alaska. Although authors point to the map of Wang et al 2006 for assigning the northern evergreen shrubs to the broadleaved evergreen

tree PFT, it is recommended to correct the map, and remove broadleaved evergreen tree type north of its known area before comparing with model simulation.

**Given the low fractional coverage of broadleaf evergreen PFT above 30 N, we agree, that it is confusing to show an incorrect distribution of this PFT based on the published Wang et al., (2006) paper. We will therefore remove Figure 9 from the manuscript.**

4) In the introduction (line 96-97), authors define using 1 degree spatial resolution as major difference between this and Melton and Arora (2016) studies. Using higher spatial resolution appears as a major motivation for the study. However there is no clear statement or conclusion on the effect of resolution on improving simulated distributions of PFT, LAI or carbon fluxes. Even if there is no significant improvement, which is possible, the result should be stated clearly.

**Although we do discuss this aspect in the Discussions section we agree that a mention of the effect of implementing the competition module at a finer resolution is not included in the abstract or the Conclusions section of the manuscript. We will include an additional table in the manuscript that will compare correlation coefficients and root mean square differences (RMSDs) for the simulated versus observed tree, grass, vegetated and bare ground fractions between the fine and coarse spatial resolution implementations of the competition module. This will allow us to quantitatively discuss any improvements (or otherwise) that we have obtained by implementing the competition module at the finer spatial resolution. We will also include the derived conclusion in the abstract and revise the manuscript to include more discussion of this important aspect.**

Minor, technical corrections Line 35. Authors point that some processes need further development. Suggest to briefly mention which actual processes need attention.

**Although we do mention lack of shrubs and moss PFTs and an inadequate representation of permafrost in the model as processes that need attention in the**

[Figure]

abstract of our manuscript, we will revise the abstract so that it is clear that it is
these processes that need further improvement in our modelling framework.

 Reviewer 2

1. Establishing reasonability: The primary objective of a model is to 'look' at the terrestrial biosphere for such conditions that cannot be observed. For example, experimental manipulations of ecosystems, simulations for climatic conditions in the future. It would be highly improbable to have model outputs exactly match observational data. Two independent observational datasets have differences. Therefore, while assessing model performance, it is essential to establish quantitatively what 'reasonable performance' is. It could be a measure of deviation from multi-observational mean (ensemble) or a threshold for spatial correlation (R2). Establishing this 'reasonability' will help users of this model to limit themselves to assessing simulation results for certain geographical regions and/or certain Plant Functional Types (PFTs).

 **The reviewer suggests "Establishing this 'reasonability' will help users of this model to limit themselves to assessing simulation results for certain geographical regions". It appears reviewer 2 is suggesting to calculate correlation coefficients for different geographical regions or mention in the manuscript that if the bias in simulated fractional coverage of a given PFT (or tree, grass or vegetated fraction) is less than some magnitude than the simulated fractional coverage can be deemed reasonable. One possible way to assign a limit on acceptable bias is use the root mean square difference (RMSD) between the two observation-based data sets themselves as a measure observational uncertainty. However, this would be incorrect since both observational-based estimates have their own uncertainties. In addition, uncertainty is introduced by the process through which 20 something vegetation types from land cover products (section 2.3.1 of our manuscript) are mapped on to nine PFTs that CLASS-CTEM uses, which we are unable to quantify. So in essence the RMSD between the two observation-based data sets is not a true measure of the uncertainty in observations and there-**

**fore cannot be used to conclusively say if model can be deemed reasonable in certain regions or not. We agree with reviewer 2's overall objective to better quantify in which regions the model performs better. However, this information is already shown using spatial plots of differences between the model simulated and observation-based fractional coverage of different PFTs. These plots also allow to visually identify areas where the model simulated fractional coverage of PFTs show bias relative to both observation-based data sets (e.g. lower vegetation fraction in the semi-arid south-west United States and higher in Arctic Canada). We are reluctant to overinterpret our simulated results by using some measure of observational uncertainty which we are unable to quantify reliably.**

2. More comprehensive model performance assessment: a) A critical element of model evaluation that is missing is comparison with other published DGVM outputs. DGVMs are similar in their basic structure but differ with respect to certain processes and/or parameterization. Thus comparison with other DGVM outputs will allow the authors to establish how the carbon cycle dynamics and/or terrestrial vegetation patterns of CLASS-CTEM compare with that of other models. This will further help to establish the 'reasonability' of CLASS-CTEM. I would suggest using the ensemble of DGVM outputs from the TRENDY project, which is a consortium of DGVM groups who set up a project to investigate further the spatial trends in Net Biome Production (NBP) and agreed to perform a factorial set of DGVM simulations over the historical period, 1901 – 2010 (see http://dgvm.ceh.ac.uk/ ). b) Since the primary objective is to compare model performance, it will be a good idea to compare the model GPP to a satellite based model output like the MODIS MOD17 data (see http://www.ntsg.umt.edu/project/MOD17/ ! 'Data Product'). Using this will also help in establishing observational uncertainty.

**Actually, CLASS-CTEM did participate in the most recent TRENDY project whose results contributed to the 2016 global carbon project (Le Quéré et al., 2016). Figure 1 (at the end of this reply) shows CLASS-CTEM simulated global net atmosphere-land $CO_2$ flux compared to other participating models and the ter-**

**restrial sink calculated as the residual of the global carbon budget. Figure 1 shows that while the CLASS-CTEM simulated net atmosphere-land CO2 flux exhibits somewhat higher inter-annual variability compared to some other models overall the model's response is broadly consistent with other models and the terrestrial sink calculated as the residual of the global carbon budget. We will make a note of this in our revised manuscript referencing the Le Quéré paper but we are reluctant to include this figure in our manuscript. The focus of our manuscript is on simulating competition between PFTs over the North American domain. Also we do compare model-simulated NBP with inversion- and other model-based estimates for North America at the end of section 3.3. CLASS-CTEM simulated GPP is compared to a an observation-based product (Beer et al., 2010) in Melton and Arora (2016). We will make a note of this in the revised manuscript.**

3. Regional analysis: The paper currently does perform a regional analysis, but it is qualitative. The future users of CLASS-CTEM would want to know in which specific regions of North America is the model performance reliable enough. I would thus suggest using shapefiles to demarcate the different regions being studied and then carry out statistical comparisons for each region. It would be helpful for the reader if the maps of individual regions are explicitly shown in the paper.

**We do not agree that our analysis is qualitative. We provide quantitative spatial plots of differences in simulated fractional coverage compared to the two observation-based data sets including spatial correlations and spatially-averaged RMSD. We also do not see benefit of using shapefiles to identify different regions as the entire model domain is visible in our plots. The existing spatial plots in Figures 3 through 8 do show differences in simulated values of fractional coverage of vegetation from the two observation-based data sets. The objective of these plots is indeed to assess the regions where the model performs well and where it does not.**

[Figure]

4. Time series comparison: According to the authors, the primary goal of the paper is to evaluate the performance of the competition module of CLASS-CTEM. While comparing current vegetation patterns and productivity with that of observational data is a good way to evaluate competition, it is not comprehensive since it looks at only one time slice and equilibrium conditions. Competition can be observed in reality when an ecosystem is subjected to environmental changes and/or disturbance. The time-series data of both CLASS-CTEM data and observational data both incorporate the response of the terrestrial ecosystems to changes in the climate and atmospheric CO2 concentrations. Thus a better evaluation of the competition module should include equivalent time series comparisons of model and observation.

**This is a valid point and this aspect of the model has been assessed in the past where the model was applied at point scale at different locations (Arora and Boer, 2006; Shrestha et al., 2016). Competition between PFTs occurs at a very slow rate and PFTs reach their equilibrium fractional coverage over decades. Observation-based data at a single location at these time scales are difficult to obtain. So published CLASS-CTEM papers (Arora and Boer, 2006; Shrestha et al., 2016) have used other information to evaluate how the model simulated fractional coverages of different PFTs evolve in time including time to equilibrium. We will make a note of this aspect and refer to these papers when revising our manuscript.**

5. Attributing deviations from observation: The authors admit that it is difficult to determine whether the model limitations are due to biogeochemical parameterizations or structural limitations (Pg 20, lines 578-580). If this is the case, then attributing deviations from observations is mere speculation. While speculation backed by scientific logic is okay for the discussions section, it shouldn't be present in the abstract since this isn't something that the authors have assessed. That being said, model limitations can be diagnosed by tuning parameters. The biogeochemical parameter that is the main suspect causing deviations from observation should be identified and parallel

simulations should be run using varying values of the parameter and/or using values which are a closer match to North American species.

**We do not attribute deviations from observations to a specific cause but rather say (page 20, lines 578-580) that it is difficult to conclusively determine whether limitations in the biogeochemistry parameterizations or the structural limitation that the model does not represent shrub, moss and lichen PFTs is the reason for model's deviations from observations. The discussion that follows then uses scientific reasoning to make the case that it is more likely the absence of the shrub, moss and lichen PFTs in the model together with an inadequate representation of permafrost that causes grasses to perform overly well.**

**We do not agree that model limitations related to biogeochemical parameterizations or model's structure can be diagnosed by tuning. The model has more than 100 parameters for each of its nine PFTs and competition between PFTs depends on about 50 of those. Optimizing model parameters without running into the problem of equifinality and while keeping the model behaviour realistic is a non-trivial exercise. Model tuning is a long-term and ongoing process and every model evaluation exercise helps to improve model parameters. However, we feel that tuning of model parameters cannot and should not be used to hide or overcome limitations in model's structure and its biogeochemical parameterizations.**

6. Impact of change in spatial scale: The authors say that one of the goals of the paper was to look at how the 1 degree version of the model changed results when compared to the earlier study performed using the 3.75 degree version (Pg 23, lines 674-678). The authors make qualitative statements and not any quantitative assessments. I was expecting to see RMSD and R2 values for both model versions. This would help establish whether a higher spatial resolution helps to improve model performance or not.

**Thank you pointing this out. This issue is also raised by reviewer 1. We will**

**include an additional table in the manuscript that will compare correlation coefficients and RMSDs for the simulated versus observed tree, grass, vegetated and bare ground fractions between the fine and coarse spatial resolution implementations of the competition module. This will allow us to quantitatively discuss any improvements (or otherwise) that we have obtained by implementing the competition module at the finer spatial resolution.**

Minor comment: The paper would be of much more value to the climate science and/or carbon cycle community if the authors used more of carbon cycle flux and/or stock terms/values to assess CLASS-CTEM performance. While it is useful to have a detailed analysis of the major PFTs, analyzing the minor PFTs doesn't add much value to the model assessment. Instead grouping PFTs under broader categories (eg. Trees/grass and/or needleleaf/broadleaf trees) adds value to the paper as it makes comparisons with other models and datasets simpler.

**The typical carbon cycle fluxes and pools simulated by the CLASS-CTEM model have been assessed at point, regional and global scales in several earlier publications. We will ensure that references to these publications are complete. We agree that analysis of minor PFTs doesn't add much value and therefore have decided to remove Figure 9 in the revised version of the manuscript.**
* * *
**References**

Arora, V. K. and Boer, G. J.: Simulating Competition and Coexistence between Plant Functional Types in a Dynamic Vegetation Model, Earth Interact., 10(10), 1–30, 2006.

Beer, C., Reichstein, M., Tomelleri, E., Ciais, P., Jung, M., Carvalhais, N., Rödenbeck, C., Arain, M. A., Baldocchi, D., Bonan, G. B., Bondeau, A., Cescatti, A., Lasslop, G., Lindroth, A., Lomas, M., Luyssaert, S., Margolis, H., Oleson, K. W., Roupsard, O., Veenendaal, E., Viovy, N., Williams, C., Woodward, F. I. and Papale, D.: Terrestrial Gross

Carbon Dioxide Uptake: Global Distribution and Covariation with Climate, Science, 329(5993), 834–838, 2010.

Le Quéré, C., Andrew, R. M., Canadell, J. G., Sitch, S., Korsbakken, J. I., Peters, G. P., Manning, A. C., Boden, T. A., Tans, P. P., Houghton, R. A., Keeling, R. F., Alin, S., Andrews, O. D., Anthoni, P., Barbero, L., Bopp, L., Chevallier, F., Chini, L. P., Ciais, P., Currie, K., Delire, C., Doney, S. C., Friedlingstein, P., Gkritzalis, T., Harris, I., Hauck, J., Haverd, V., Hoppema, M., Klein Goldewijk, K., Jain, A. K., Kato, E., Körtzinger, A., Landschützer, P., Lefèvre, N., Lenton, A., Lienert, S., Lombardozzi, D., Melton, J. R., Metzl, N., Millero, F., Monteiro, P. M. S., Munro, D. R., Nabel, J. E. M. S., Nakaoka, S.-I., O'Brien, K., Olsen, A., Omar, A. M., Ono, T., Pierrot, D., Poulter, B., Rödenbeck, C., Salisbury, J., Schuster, U., Schwinger, J., Séférian, R., Skjelvan, I., Stocker, B. D., Sutton, A. J., Takahashi, T., Tian, H., Tilbrook, B., van der Laan-Luijkx, I. T., van der Werf, G. R., Viovy, N., Walker, A. P., Wiltshire, A. J. and Zaehle, S.: Global Carbon Budget 2016, Earth Syst Sci Data, 8(2), 605–649, doi:10.5194/essd-8-605-2016, 2016.

Melton, J. R. and Arora, V. K.: Competition between plant functional types in the Canadian Terrestrial Ecosystem Model (CTEM) v. 2.0, Geosci Model Dev, 9(1), 323–361, doi:10.5194/gmd-9-323-2016, 2016.

Shrestha, R. K., Arora, V. K. and Melton, J. R.: The sensitivity of simulated competition between different plant functional types to subgrid-scale representation of vegetation in a land surface model, J. Geophys. Res. Biogeosciences, 121(3), 809–828, doi:10.1002/2015JG003234, 2016.

Wang, A., Price, D. T. and Arora, V.: Estimating changes in global vegetation cover (1850–2100) for use in climate models, Glob. Biogeochem. Cycles, 20(3), n/a-n/a, doi:10.1029/2005GB002514, 2006.

Wu, Y., Verseghy, D. L. and Melton, J. R.: Integrating peatlands into the coupled Canadian Land Surface Scheme (CLASS) v3.6 and the Canadian Terrestrial Ecosystem Model (CTEM) v2.0, Geosci Model Dev, 9(8), 2639–2663, doi:10.5194/gmd-9-2639-

2016, 2016.
* * *
[Figure]

Figure 1: Comparison of CLASS-CTEM simulated net atmosphere-land $CO_2$ flux to values simulated by other participating models in the TRENDY project (from year 2016) and the terrestrial sink which is calculated as the residual of the global carbon budget for the period 1959-2015.

**Fig. 1.**

---

## Author Comment (AC2) · 22 Jun 2017

Response to reviewer 2's comments included together with the response to reviewer 1 comments

---

## Referee Report (RR1)

I have gone through the authors' response to the reviewer comments as well as the revised paper. I appreciate the changes made in response to the comments. There have been areas where the authors have argued against changes recommended. I find the arguments reasonable and hence agree with authors' point of view. The paper now is a much more comprehensive evaluation of competition module of the CLASS-CTEM modelling framework with clear assessment of limitations. I would thus recommend publishing this paper as it is.